# A Ship Discrimination Method Based on High-Frequency Electromagnetic Theory

Yaomin He [1,2], Huizhang Yang [1], Huafeng He [2], Junjun Yin [3] and Jian Yang [1,*]

1 Department of Electronic Engineering, Tsinghua University, Beijing 100084, China; heym20@mails.tsinghua.edu.cn (Y.H.); hzyang@njust.edu.cn (H.Y.)
2 College of Missile Engineering, Rocket Force Engineering University, Xi'an 710025, China; he1071936827@sina.cn
3 School of Computer and Communication Engineering, University of Science and Technology Beijing, Beijing 100083, China; junjun-yin@ustb.edu.cn
* Correspondence: yangjian-ee@tsinghua.edu.cn; Tel.: +86-6277-0317

**Abstract:** Ship target detection using radar has important applications in the military and civilian fields. As a decoy, the corner reflector (CR) can successfully deceive a radar by its strong radar cross-section (RCS) to protect a ship. In order to discriminate between a CR and ship, this paper proposes a discrimination method based on three-dimensional characteristics. First, we deduce the basic scattering of CR by the high-frequency electromagnetic theory, and propose a CR decomposition which can solve the problem that the Krogager decomposition has terrible errors in clutter. Then, we introduce the definition of the main scattering polarization and give the multi-dimensional characteristic of CR. Subsequently, we analyze the spatial-time characteristic of radar based on the three-dimensional proportional guidance. With the CR mean square error (MSE), a CR discrimination method is proposed based on the time-spatial-polarization (TSP) joint domains. Finally, the proposed method is analyzed and compared using the fully polarimetric data of Feko software, which can achieve 95% discrimination probability and 4.1% false alarm probability.

**Keywords:** polarimetric radar; ship discrimination; corner reflector; electromagnetic theory





## 1. Introduction

Ship target detection is essential for many applications in marine target monitoring [1], maritime rescue [2], and port security [3]. Especially in the military field, the ability to effectively detect a ship target whilst being interfered with by false targets is very important. The corner reflector (CR) can be used for polarimetric synthetic aperture radar (SAR) calibration [4], false targets [5], and polarimetric radar interferometer [6] due to many advantages [7]. In the application of a false target, CR can successfully deceive a radar by its strong radar cross-section (RCS) for protecting important targets, i.e., ships. In this paper, our goal was to discriminate the CR and ship for target detection. In the existing literature, there are usually three types of discrimination methods, i.e., high-range resolution profile (HRRP), micro-Doppler, and polarization, as shown in the following.

For the discrimination method based on HRRP, the difference between the scattering points between the ship and single CR was analyzed in [8], and then the support vector machine (SVM) was utilized for discrimination. Subsequently, to discriminate the array CRs, six parameters of HRRP were introduced, which could find that the CR HRRP was easily affected by the arrangement and spacing in [9]. A novel HRRP-based method by changing the modulation waveform of a transmitted wave was proposed in [10]. The discriminations of the dilute jamming and centroid jamming of CR were analyzed in [11], which ignored the relationship between the number of sub-strong scattering points and the detection threshold. After extracting multiple geometric characteristics of HRRP, many references also proposed different discrimination methods based on data mining. For example, the sparse dictionary

learning was utilized to reconstruct the target, and then a discrimination method based on the error ratio of ships and CRs was proposed in [12]. Nine parameters of HRRP were introduced for training the deep neural network in [13]. The extreme learning machine algorithm was utilized in [14]. Although the difference between the ship and single CR is obvious, the array CR can realistically simulate a ship's radial distribution through a reasonable arrangement, so that there is no significant difference between the array CR and ship in HRRP [15]. Additionally, the polarization angle [16], RCS [17], and HRRP [18] all have very similar characteristics to ships, by optimizing the arrangement, spacing, and type of array CRs. The above conclusions can effectively improve the similarity between the CR and ship's HRRP, thus reducing the effectiveness of the HRRP-based methods.

The second type of discrimination is based on the micro-Doppler. Chen first introduced the concept of the micro-Doppler, which could be used in many different applications in [19,20], e.g., the vehicle discrimination method [21] and the false warhead discrimination method [22]. Subsequently, some references also proposed the micro-Doppler-based methods to discriminate CR and ship. For example, the micro-motion model of CR with the effect of ocean waves was established, and the difference in roll direction was utilized to discriminate in [23,24]. According to that, the mass and volume of CR were much smaller than that of the ship, and it could find that the CR micro-Doppler phenomenon was more obvious in [25]. Zhu et al. expanded the micro-Doppler characteristic with three parameters in [26]. In summary, the aforementioned references all analyzed the difference between a CR and ship in micro-Doppler. However, these assumed that the ocean wave only propagates in one fixed direction, and ignored the influence of wind speed or relative motion in [23]. Furthermore, the micro-motion model of a ship should include 6 degrees of freedom, but there are only two degrees of freedom analyzed to simplify the model in [23–26].

Last but not least, there are lots of discrimination methods based on the polarization information. The Krogager decomposition was used to decompose the polarization scattering matrix (PSM) of ships and CRs, and then the changes of basic scattering were compared in [27]. The Krogager decomposition was also utilized to obtain the coefficient of each basic scattering, and a discrimination method by SVM was proposed in [15]. Fang et al. optimized the odd scattering and even scattering matrices, and then utilized the polarimetric similarity parameter to discriminate CR in [28,29]. Wang et al. introduced five polarization invariants and one polarization shape factor of CR, and then proposed an SVM-based method to discriminate CR in [30,31]. Considering the measurement accuracy of the polarization phase in a sea clutter environment, two polarization invariants were ignored, and thus were not used as training parameters for SVM in [32]. Similarly, four characteristics of HRRP and two characteristics of polarization were analyzed in [33]. In brief, the aforementioned references based on polarization could be divided into two categories. (1) By extracting the polarization characteristics of the CR and ship, the discrimination methods are proposed by SVM. Such methods are not only susceptible to the data's authenticity, but also to the arrangement of array CRs. (2) The coefficient of each basic scattering is obtained based on polarization decomposition, which could be utilized to discriminate the ship and CR. The theoretical basis of a secondary category is that CR is mainly odd scattering, whilst that of the ship is mainly even scattering [27,28]. However, the CR might mainly be even scattering and the ship might also have odd scattering, e.g., when the incident wave is far from the axis of symmetry. Therefore, the discrimination method would have the problem of high false alarm probability, which is based solely on the coefficient of each basic scattering.

The main goal of this paper was to discriminate the CR and ship. In contrast to the above three types of methods (i.e., HRRP, micro-Doppler, and polarization), we will analyze the scattering characteristics of the CR and ship based on high-frequency electromagnetic theory, which can obtain the characteristics of the time–spatial–polarization joint domains of the target. Regarding the analysis of the target characteristics using high-frequency electromagnetic theory, scholars have performed much pioneering work. Aiming at the

CR formed by many surfaces on ships, Reference [34] developed a criterion that gave the required angle as a function of the desired RCS reduction and the electrical size of CR.

In [35], the PO and physical theory of diffraction (PTD) were used to determine the RCS of dihedral CRs in the azimuthal plane. In contrast to the method of PTD in [35], the uniform theory of diffraction (UTD) plus an imposed edge diffraction extension was used to predict the RCS of dihedral CRs in [36]. In comparison to shooting-bouncing-ray (SBR) with measurements and other rigorous theoretical methods, the authors in [37] verified that SBR yielded good results for calculating RCS from a three-dimensional target.

In order to analyze the PSM of a target, the authors in [38] proposed the analytic physical optics method based on PO-geometrical optics (GO). Since numerical methods cannot be applied to some complex structures, an asymptotic method based on the PO combined with the GO approximations was proposed for octahedral reflectors and icosahedral CRs in [39]. In [40], an analytic scattering model for 3D bistatic scattering was derived from a dihedral CR using GO and PO. Compared with the aforementioned high-frequency electromagnetic methods, we proposed a joint method of PO-GO-SBR in [41], which could derive a closed-form electromagnetic formula for a target with any structure. Based on the high-frequency electromagnetic theory, the present method in this paper independently analyzes the single reflection (SR), double reflection (DR), and triple reflection (TR) of PSM by controlling the number of scattering in the SBR, which is important in the wake of obtaining the odd and even scattering characteristics of target. In order to discriminate between the CR and ship, the polarization characteristic and the spatial–time characteristic of a target were developed. The main innovations of this paper are as follows.

1. The CR decomposition based on the main polarization. For the basic scattering of CR, the Krogager decomposition [27–29] has three assumptions, as presented in Section 3, which would increase the coefficient errors of odd scattering and even scattering as SCR decreases. To cure the above problem, this paper deduces the SR, DR, and TR of CR with the electromagnetic theory, and proposes a novel CR decomposition based on the main polarization, which could accurately obtain the coefficients of odd scattering and even scattering in clutter.

2. The multi-dimensional characteristic of CR. Aiming at the one-dimensional characteristic of CR and the ship, the characteristics of HRRP [8–14], micro-Doppler [23–26], polarization [27–32] are separately analyzed. For the multi-dimensional characteristic, only the authors in [33] comprehensively analyzed HRRP and polarization. However, its multi-dimensional characteristic is only a superposition of characteristic quantities, and does not consider the interaction between characteristics. Therefore, this paper introduces the definition of the main scattering polarization (MSP), and gives the multi-dimensional characteristic of CR based on the polarization and angle, i.e., the amplitude of MSP is a single-peak curve when the azimuth or pitch angle of an incident wave is monotonic within $35°$ without clutter. Based on the above property, the multi-characteristic between the ship and CR would not be affected by the angle of the incident wave, which is more applicable.

3. The CR discrimination based on the TSP joint domains. In contrast to References [8–33,42], we focus on analyzing the characteristic changes of a target during the radar movement, rather than being limited to a fixed pulse. That is, this paper utilizes multiple pulses for discrimination. Based on the three-dimensional proportional guidance method with the angle constraint, this paper gives the spatial-time characteristic of the radar, i.e., the azimuth and pitch angles of a radar's line of sight (LOS) change monotonically as the radar approaches the ship. Subsequently, we introduce a definition of the CR-MSE parameter, and propose a novel CR discrimination method based on the TSP joint domains, which can achieve 95% discrimination probability and a 4.1% false alarm probability.

This paper is organized as follows. Section 2 analyzes the scattering model of the CR: Section 3 proposes a CR decomposition; Section 4 proposes a CR discrimination method

based on TSP joint domains; Section 5 performs a mathematical simulation; and Section 6 concludes this paper.

## 2. The Scattering Model of CR

For the scattering model, the numerical algorithms usually have high accuracy, i.e., the method of moment (MoM) [43], and finite difference time domain (FDTD) [44], which request huge computational complexity. Herein, the radar frequency belongs to a higher frequency (i.e., X, Ku, K, Ka), and the size of CR and resolution are greater than 10 times the wavelength. Therefore, the high-frequency approximate algorithms, i.e., PO, GO, and SBR, have the advantages of high accuracy and computational complexity at the same time, which are suitable for analyzing the scattering model of CR [40]. In this paper, we will use the high-frequency approximate algorithms to obtain the scattering model of CR, and then analyze the polarization characteristic of CR, as shown below.

There is a target $V$ which is a distance away from the target observation point $O$. The target center is $O'$, and the $\overrightarrow{O'O}$ vector is **r**. The incident wave source is $Q$, and the $\overrightarrow{QO'}$ vector is **s**. We randomly chose a triangular facet $ABC$ on the target surface, where its normal vector is $\hat{\mathbf{n}}$. The vector from $O'$ to the center of $ABC$ is **r**′. The electric field and magnetic field of the incident wave are $\mathbf{E}_i$, $\mathbf{H}_i$, respectively, and the electric field and magnetic field of the reflected wave are $\mathbf{E}_r$, $\mathbf{H}_r$, respectively. The scattering relationship is displayed in Figure 1.

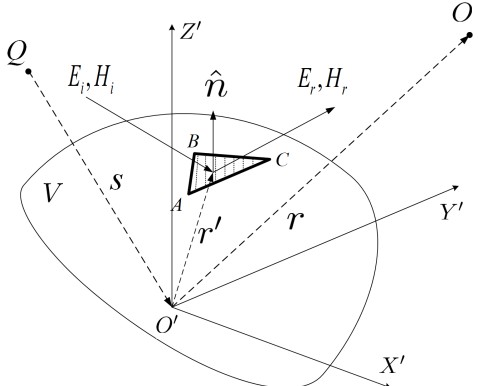

**Figure 1.** The relationship between the incident wave, reflected wave, and target.

The relationship between the incident wave and reflected wave can be obtained through the Stratton–Chu equation, i.e.,

$$\mathbf{E}_r(\mathbf{r}) = \int\int [-j\omega\mu(\hat{\mathbf{n}} \times \mathbf{H})G(\mathbf{r},\mathbf{r}') + (\hat{\mathbf{n}} \times \mathbf{E}) \times \nabla'G(\mathbf{r},\mathbf{r}') + (\hat{\mathbf{n}} \cdot \mathbf{E})\nabla'G(\mathbf{r},\mathbf{r}')]ds'$$

$$\mathbf{H}_r(\mathbf{r}) = \int\int [j\omega\varepsilon(\hat{\mathbf{n}} \times \mathbf{E})G(\mathbf{r},\mathbf{r}') + (\hat{\mathbf{n}} \times \mathbf{H}) \times \nabla'G(\mathbf{r},\mathbf{r}') + (\hat{\mathbf{n}} \cdot \mathbf{H})\nabla'G(\mathbf{r},\mathbf{r}')]ds' \tag{1}$$

where $\omega$ is the angular frequency, $\varepsilon$ is the dielectric constant of the propagation medium, $\mu$ is the magnetic permeability, and $s'$ is the area of $ABC$. **E** is the total electric field, i.e., $\mathbf{E} = \mathbf{E}_i + \mathbf{E}_r$. **H** is the total magnetic field, i.e., $\mathbf{H} = \mathbf{H}_i + \mathbf{H}_r$. $G(\mathbf{r},\mathbf{r}')$ is the Green's function as follows.

$$G(\mathbf{r},\mathbf{r}') = \frac{e^{-jk|\mathbf{r}-\mathbf{r}'|}}{4\pi|\mathbf{r}-\mathbf{r}'|}, \tag{2}$$

where $k = 2\pi/\lambda$.

Subsequently, we can simplify the reflected wave's electric field $\mathbf{E}_r$ by the Huygens principle, viz.,

$$\mathbf{E}_r(\mathbf{r}) = \nabla \times \int\int (\hat{\mathbf{n}} \times \mathbf{E})G(\mathbf{r},\mathbf{r}')ds' + \frac{1}{j\omega\varepsilon}\nabla \times \nabla \times \int\int G(\mathbf{r},\mathbf{r}')(\hat{\mathbf{n}} \times \mathbf{H})ds' . \tag{3}$$

In the same way, the reflected wave's magnetic field $\mathbf{H}_r$ is

$$\mathbf{H}_r(\mathbf{r}) = \nabla \times \int \int (\widehat{\mathbf{n}} \times \mathbf{H}) G(\mathbf{r}, \mathbf{r}') ds' - \frac{1}{j\omega\mu} \nabla \times \nabla \times \int \int G(\mathbf{r}, \mathbf{r}')(\widehat{\mathbf{n}} \times \mathbf{E}) ds' \quad . \tag{4}$$

Since the integral terms in Equations (3) and (4) are difficult to perform, we utilize high-frequency approximation methods for simplification.

### 2.1. Single Reflection

For the SR of CR, its electromagnetic scattering is analyzed using the Physical Optics (PO) [45]. First, the distance $D_R$ between $O$ and $ABC$ in Figure 1 is

$$D_R = |\mathbf{r} - \mathbf{r}'| = |\mathbf{r}| \left( 1 - \frac{\mathbf{r} \cdot \mathbf{r}'}{|\mathbf{r}|^2} + \frac{r'^2}{2|\mathbf{r}|^2} - \frac{(\mathbf{r} \cdot \mathbf{r}')^2}{2|\mathbf{r}|^4} + \dots \right) \approx |\mathbf{r}| - \widehat{\mathbf{r}} \cdot \mathbf{r}' \quad , \tag{5}$$

where $O$ is far away from $O'$, i.e., $|\mathbf{r}| >> |\mathbf{r}'|$, and $\widehat{\mathbf{r}} = \mathbf{r}/|\mathbf{r}|$.

In the same way, the distance $D_S$ between $Q$ and $ABC$ is

$$D_S = |\mathbf{s} + \mathbf{r}'| \approx |\mathbf{s}| + \widehat{\mathbf{s}} \cdot \mathbf{r}' \quad , \tag{6}$$

where $\widehat{\mathbf{s}} = \mathbf{s}/|\mathbf{s}|$.

Then, Green's function is simplified by Equation (5), i.e.,

$$\begin{aligned} G(\mathbf{r}, \mathbf{r}') &= \frac{\exp(-jk|\mathbf{r}-\mathbf{r}'|)}{4\pi|\mathbf{r}-\mathbf{r}'|} = \frac{\exp(-jk|\mathbf{r}|)\exp(jk\widehat{\mathbf{r}}\cdot\mathbf{r}')}{4\pi(|\mathbf{r}|-\widehat{\mathbf{r}}\cdot\mathbf{r}')} \\ &= \frac{\exp(-jk|\mathbf{r}|)\exp(jk\widehat{\mathbf{r}}\cdot\mathbf{r}')}{4\pi} \frac{1}{|\mathbf{r}|}\left(1 + \widehat{\mathbf{r}} \cdot \frac{\mathbf{r}'}{|\mathbf{r}|} + \dots\right) \approx \frac{\exp(-jk|\mathbf{r}|)\exp(jk\widehat{\mathbf{r}}\cdot\mathbf{r}')}{4\pi|\mathbf{r}|} \end{aligned} \quad . \tag{7}$$

Therefore, $\mathbf{E}_r$ can be further simplified [46], i.e.,

$$\mathbf{E}_r(\mathbf{r}) = jk\frac{e^{-jk|\mathbf{r}|}}{4\pi|\mathbf{r}|} \int \int \left\{ \sqrt{\frac{\mu}{\varepsilon}}[\widehat{\mathbf{r}} \times (\widehat{\mathbf{r}} \times (\widehat{\mathbf{n}} \times \mathbf{H}))] - \widehat{\mathbf{r}} \times (\widehat{\mathbf{n}} \times \mathbf{E}) \right\} e^{jk\widehat{\mathbf{r}}\cdot\mathbf{r}'} ds' \quad . \tag{8}$$

On a perfect electric conductor, the area can be approximated as a plane. Then, the property is exhibited in Equation (9) and therefore the radius of curvature on the target surface and the nearby surface is much larger than the wavelength.

$$\begin{cases} \widehat{\mathbf{n}} \times \mathbf{E} = 0 \\ \widehat{\mathbf{n}} \times \mathbf{H} = 2\widehat{\mathbf{n}} \times \mathbf{H}_i \end{cases} \quad . \tag{9}$$

Subsequently, $\mathbf{E}_r$ can be obtained using Equations (8) and (9), viz.,

$$\mathbf{E}_r(\mathbf{r}) = jk\frac{e^{-jk|\mathbf{r}|}}{2\pi|\mathbf{r}|} \int \int \left\{ \sqrt{\frac{\mu}{\varepsilon}}[\widehat{\mathbf{r}} \times (\widehat{\mathbf{r}} \times (\widehat{\mathbf{n}} \times \mathbf{H}_i))] \right\} e^{jk\widehat{\mathbf{r}}\cdot\mathbf{r}'} ds', \tag{10}$$

where $\mathbf{H}_i$ is the incident wave's magnetic field, i.e., $\mathbf{H}_i = \mathbf{H}_0 e^{j(-k \cdot D_S + 2\pi f_0 t)}/D_S$. Furthermore, $\mathbf{H}_i$ can be simplified by the far-field approximation in Equation (11).

$$\mathbf{H}_i = \frac{\mathbf{H}_0}{|\mathbf{s}|} e^{j(-k|\mathbf{s}|+2\pi f_0 t)} e^{j(-k\widehat{\mathbf{s}}\cdot\mathbf{r}')} . \tag{11}$$

Then, let the substitute Equation (11) into (10), where $\mathbf{E}_r$ is

$$\mathbf{E}_r(\mathbf{r}) = jk\frac{e^{-jk(|\mathbf{r}|+|\mathbf{s}|)+j2\pi f_0 t}}{2\pi|\mathbf{r}||\mathbf{s}|} \sqrt{\frac{\mu}{\varepsilon}}[\widehat{\mathbf{r}} \times (\widehat{\mathbf{r}} \times (\widehat{\mathbf{n}} \times \mathbf{H}_0))] \int \int e^{jk(\widehat{\mathbf{r}}-\widehat{\mathbf{s}})\cdot\mathbf{r}'} ds' \quad . \tag{12}$$

The properties of the plane electromagnetic waves are

$$
\begin{cases}
\mathbf{H}_0 = \sqrt{\frac{\varepsilon}{\mu}}\widehat{\mathbf{s}} \times \mathbf{E}_0 \\[2mm]
\mathbf{E}_0 = -\sqrt{\frac{\mu}{\varepsilon}}\widehat{\mathbf{s}} \times \mathbf{H}_0
\end{cases}.
\tag{13}
$$

Therefore, $\mathbf{E}_r$ can be further simplified in Equation (14).

$$
\mathbf{E}_r(\mathbf{r}) = jk\frac{e^{-jk(|\mathbf{r}|+|\mathbf{s}|)+j2\pi f_0 t}}{2\pi|\mathbf{r}||\mathbf{s}|}I\{(\widehat{\mathbf{n}}\cdot\mathbf{E}_0)[(\widehat{\mathbf{r}}\cdot\widehat{\mathbf{s}})\widehat{\mathbf{r}}-\widehat{\mathbf{s}}] - (\widehat{\mathbf{n}}\cdot\widehat{\mathbf{s}})[(\widehat{\mathbf{r}}\cdot\mathbf{E}_0)\widehat{\mathbf{r}}-\mathbf{E}_0]\},
\tag{14}
$$

where the integral part $I$ [47] is

$$
\begin{aligned}
I &= \int\int e^{jk(\widehat{\mathbf{r}}-\widehat{\mathbf{s}})\cdot\mathbf{r}'}ds' \\[2mm]
&= \frac{1}{jk\{[(\widehat{\mathbf{r}}-\widehat{\mathbf{s}})\cdot\widehat{\mathbf{n}}]^2-|\widehat{\mathbf{r}}-\widehat{\mathbf{s}}|^2\}}\cdot\sum_{m=1}^{M_I}\{(\widehat{\mathbf{r}}-\widehat{\mathbf{s}})\times\widehat{\mathbf{n}}\cdot(\mathbf{a}_m-\mathbf{a}_{m-1}) \\[2mm]
&\quad\cdot\exp\left[j\frac{k}{2}(\widehat{\mathbf{r}}-\widehat{\mathbf{s}})\cdot(\mathbf{a}_m+\mathbf{a}_{m-1})\right]\cdot\frac{\sin\left[\frac{k}{2}(\widehat{\mathbf{r}}-\widehat{\mathbf{s}})\cdot(\mathbf{a}_m-\mathbf{a}_{m-1})\right]}{\frac{k}{2}(\widehat{\mathbf{r}}-\widehat{\mathbf{s}})\cdot(\mathbf{a}_m-\mathbf{a}_{m-1})}\}
\end{aligned}
\tag{15}
$$

where $M_I$ is determined according to the number of sides of $ABC$, i.e., $M_I = 3$. $\mathbf{a}_m$ is the vertice of $ABC$, $m = 1, 2, \ldots, M_I$.

In brief, Equation (14) is the electric field of a bistatic radar. In the case of a monostatic radar, $\mathbf{r} = -\mathbf{s}$, and the property holds, i.e., $\widehat{\mathbf{r}}\cdot\mathbf{E}_0 = -\widehat{\mathbf{s}}\cdot\mathbf{E}_0 = 0$. Therefore, the SR of CR can be obtained by simplifying Equation (14), as displayed in Equation (16).

$$
\mathbf{E}_{\text{single}}(\mathbf{r}) = -jk\frac{e^{j2\pi f_0(t-2|\mathbf{r}|/c)}}{2\pi|\mathbf{r}|^2}I_{\text{single}}[(\widehat{\mathbf{n}}\cdot\widehat{\mathbf{r}})\mathbf{E}_0],
\tag{16}
$$

where $c$ is the light velocity and the integral part $I_{\text{single}}$ is

$$
I_{\text{single}} = \frac{j(\widehat{\mathbf{r}}\times\widehat{\mathbf{n}})}{2k[1-(\widehat{\mathbf{r}}\cdot\widehat{\mathbf{n}})^2]}\cdot\sum_{m=1}^{M}\{(\mathbf{a}_m-\mathbf{a}_{m-1})\exp[jk\widehat{\mathbf{r}}\cdot(\mathbf{a}_m+\mathbf{a}_{m-1})]\cdot\frac{\sin[k\widehat{\mathbf{r}}\cdot(\mathbf{a}_m-\mathbf{a}_{m-1})]}{k\widehat{\mathbf{r}}\cdot(\mathbf{a}_m-\mathbf{a}_{m-1})}\}.
\tag{17}
$$

### 2.2. Double Reflection

In this section, we combine PO and SBR [48] together, which can separately analyze DR and TR for the CR of any shape. This method based on PO-SBR contains two steps, i.e., ray tracing and electromagnetic calculation. In the process of ray tracing, the CR is decomposed into a series of triangular facets, where ray tracing is performed on the triangular facets. Let $A_2$ be the triangular facet of DR, and $A_1$ be the triangular facet of SR. When the incident wave with the unit vector $\widehat{\mathbf{s}_0}$ illuminates the triangular facet $A_1$ with the normal vector $\widehat{\mathbf{n}_1}$, the unit vector of the reflected wave is $\widehat{\mathbf{r}_0}$, as presented in Figure 2. In addition, $a_s, a_r$ are the incident and reflected angles of $A_1$, respectively.

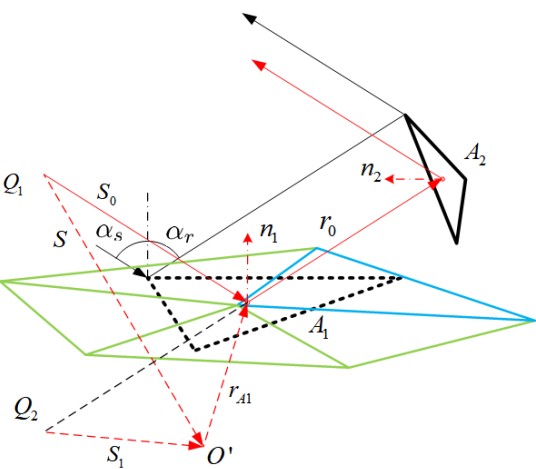

**Figure 2.** Double scattering of triangular facets.

First, Equation (18) can be given according to specular reflection.

$$\widehat{\mathbf{r}_0} \times \widehat{\mathbf{n}_1} = \widehat{\mathbf{s}_0} \times \widehat{\mathbf{n}_1}. \tag{18}$$

By multiplying Equation (18) by $(\widehat{\mathbf{r}_0} + \widehat{\mathbf{s}_0})$ and $\widehat{\mathbf{n}_1}$, respectively, the properties of $\widehat{\mathbf{r}_0}$ and $\widehat{\mathbf{s}_0}$ are

$$\begin{cases} \widehat{\mathbf{r}_0} = \widehat{\mathbf{s}_0} - 2(\widehat{\mathbf{s}_0} \cdot \widehat{\mathbf{n}_1})\widehat{\mathbf{n}_1} \\ \widehat{\mathbf{s}_0} = \widehat{\mathbf{r}_0} - 2(\widehat{\mathbf{r}_0} \cdot \widehat{\mathbf{n}_1})\widehat{\mathbf{n}_1} \end{cases}. \tag{19}$$

Then, by substituting Equations (19) into (14), $\mathbf{E}_1(\mathbf{r}_0)$ is

$$\mathbf{E}_1(\mathbf{r}_0) = jk\frac{e^{-jk(|\mathbf{r}_0|+|\mathbf{s}_0|)+j2\pi f_0 t}}{2\pi|\mathbf{r}_0||\mathbf{s}_0|}I$$
$$\cdot[-2(\widehat{\mathbf{r}_0} \cdot \widehat{\mathbf{n}_1})^2(\mathbf{E}_0 \cdot \widehat{\mathbf{n}_1})\widehat{\mathbf{r}_0} + 2(\widehat{\mathbf{r}_0} \cdot \widehat{\mathbf{n}_1})(\mathbf{E}_0 \cdot \widehat{\mathbf{n}_1})\widehat{\mathbf{n}_1} + (\widehat{\mathbf{r}_0} \cdot \widehat{\mathbf{n}_1})(\widehat{\mathbf{r}_0} \cdot \mathbf{E}_0)\widehat{\mathbf{r}_0} - (\widehat{\mathbf{r}_0} \cdot \widehat{\mathbf{n}_1})\mathbf{E}_0]. \tag{20}$$

Since the incident wave is a plane wave, i.e., $\widehat{\mathbf{s}} \cdot \mathbf{E}_0 = 0$, $\mathbf{E}_1(\mathbf{r}_0)$ can be further simplified in Equation (21).

$$\mathbf{E}_1(\mathbf{r}_0) = jk\frac{e^{-jk(|\mathbf{r}_0|+|\mathbf{s}_0|)+j2\pi f_0 t}}{2\pi|\mathbf{r}_0||\mathbf{s}_0|}I(\widehat{\mathbf{r}_0} \cdot \widehat{\mathbf{n}_1})\{2(\mathbf{E}_0 \cdot \widehat{\mathbf{n}_1})\widehat{\mathbf{n}_1} - \mathbf{E}_0\}. \tag{21}$$

If the phase part $je^{-jk(|\mathbf{r}_0|+|\mathbf{s}_0|)+j2\pi f_0 t}$ and the amplitude part $kI(\widehat{\mathbf{r}_0} \cdot \widehat{\mathbf{n}_1})/2\pi|\mathbf{r}_0||\mathbf{s}_0|$ are ignored in Equation (21), the polarization information after SR is

$$\mathbf{E}_1 = 2(\mathbf{E}_0 \cdot \widehat{\mathbf{n}_1})\widehat{\mathbf{n}_1} - \mathbf{E}_0. \tag{22}$$

In Figure 2, $A_1$ is the triangular facet of SR, and $Q_1$ is the incident wave source. Based on the equivalent image source method, we let $Q_2$ be the mirror image source. Then, $Q_2$ can be regarded as the incident wave source of DR, and $s_1$ can also be regarded as the incident source to the target center. Using the geometric relationship, we find that the vector from $Q_2$ to the center of $A_1$ and the vector from the center of $A_1$ to the center of $A_2$ are the same. Therefore, $\mathbf{s} + \mathbf{r}_{A_1} = \mathbf{s}_0$, and $\mathbf{s}_1 + \mathbf{r}_{A_1} = \mathbf{r}_0$. According to Equation (19), $\mathbf{s}_1$ after the SR is

$$\mathbf{s}_1 = \mathbf{s} - 2[(\mathbf{s} + \mathbf{r}_{A_1}) \cdot \widehat{\mathbf{n}_1}]\widehat{\mathbf{n}_1}. \tag{23}$$

Subsequently, multiply the left and right sides of Equation (23) by $(\mathbf{I} - 2\widehat{\mathbf{n}_1}\widehat{\mathbf{n}_1}^T)$, and $|\mathbf{s}_1|$ can be obtained by utilizing $|\widehat{\mathbf{n}_1}| = 1$, i.e.,

$$|\mathbf{s}_1| = |\mathbf{s} - 2(\mathbf{E}_I - 2\widehat{\mathbf{n}_1}\widehat{\mathbf{n}_1}^T)\widehat{\mathbf{n}_1}\widehat{\mathbf{n}_1}^T\mathbf{r}_{A_1}| \approx |\mathbf{s}| - 2\widehat{\mathbf{s}} \cdot (\mathbf{E}_I - 2\widehat{\mathbf{n}_1}\widehat{\mathbf{n}_1}^T)\widehat{\mathbf{n}_1}\widehat{\mathbf{n}_1}^T\mathbf{r}_{A_1}, \tag{24}$$

where $\mathbf{E}_I$ is unit matrix, and the unit vector of $\mathbf{s}_1$ is

$$\widehat{\mathbf{s}_1} = \frac{\mathbf{s}_1}{|\mathbf{s}_1|} \approx \frac{(\mathbf{E}_I - 2\widehat{\mathbf{n}_1}\widehat{\mathbf{n}_1}^T)^{-1}\mathbf{s}}{|\mathbf{s}|} = (\mathbf{E}_I - 2\widehat{\mathbf{n}_1}\widehat{\mathbf{n}_1}^T)\widehat{\mathbf{s}} \cdot \tag{25}$$

Finally, the DR of CR can be obtained by substituting Equations (22), (24) and (25) into (14), i.e.,

$$\begin{aligned} \mathbf{E}_{\text{double}}(\mathbf{r}) = jk\frac{e^{-jk(|\mathbf{r}|+|\mathbf{s}|)+j2\pi f_0 t}}{2\pi|\mathbf{r}||\mathbf{s}|} \cdot I_{\text{double}} e^{2jk\widehat{\mathbf{s}}\cdot(\mathbf{E}_I - 2\widehat{\mathbf{n}_1}\widehat{\mathbf{n}_1}^T)\widehat{\mathbf{n}_1}\widehat{\mathbf{n}_1}^T\mathbf{r}_{A_1}} \\ \cdot\{(\widehat{\mathbf{n}_2}\cdot\mathbf{E}_1)[(\widehat{\mathbf{r}}\cdot\widehat{\mathbf{s}}_1)\widehat{\mathbf{r}} - \widehat{\mathbf{s}}_1] - (\widehat{\mathbf{n}_2}\cdot\widehat{\mathbf{s}}_1)[(\widehat{\mathbf{r}}\cdot\mathbf{E}_1)\widehat{\mathbf{r}} - \mathbf{E}_1]\} \end{aligned} \tag{26}$$

where $\widehat{\mathbf{n}_2}$ is the unit normal vector of the DR's triangle facet, and $I_{\text{double}} = \int\int e^{jk(\widehat{\mathbf{r}}-\widehat{\mathbf{s}}_1)\cdot\mathbf{r}'}ds'$.

### 2.3. Triple Reflection

Similarly to DR, let $A_1$ be the triangular facet of SR with the normal vector $\mathbf{n}_1$, where the incident wave and reflected wave are $\mathbf{s}$ and $\mathbf{s}_1$, respectively. The triangular facet of DR is $A_2$ with the normal vector $\mathbf{n}_2$, where the incident wave and reflected wave are $\mathbf{s}_1$ and $\mathbf{s}_2$, respectively, as exhibited in Figure 3.

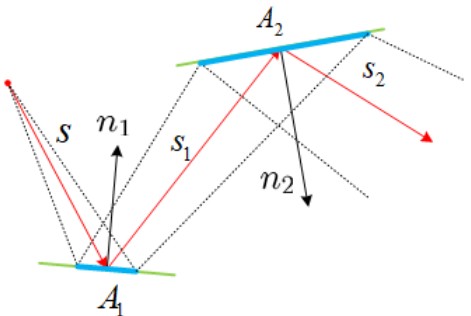

**Figure 3.** Triple scattering of triangular facets.

First, the polarization information of DR is displayed in Equation (27) by referring to Equation (22).

$$\mathbf{E}_2 = 2(\mathbf{E}_1 \cdot \widehat{\mathbf{n}_2})\widehat{\mathbf{n}_2} - \mathbf{E}_1. \tag{27}$$

Similarly referring to Equation (24), $\mathbf{s}_2$ after the DR is

$$|\mathbf{s}_2| \approx |\mathbf{s}| - 2\widehat{\mathbf{s}} \cdot \sum_{m=1}^{2}\prod_{i=1}^{m}(\mathbf{I} - 2\widehat{\mathbf{n}}_i\widehat{\mathbf{n}}_i^T)\widehat{\mathbf{n}}_m\widehat{\mathbf{n}}_m^T\mathbf{r}_{A_m} \cdot \tag{28}$$

Subsequently, the unit vector of $\mathbf{s}_2$ is

$$\widehat{\mathbf{s}_2} = \frac{\mathbf{s}_2}{|\mathbf{s}_2|} \approx \frac{\left[\prod_{i=1}^{2}(\mathbf{I} - 2\widehat{\mathbf{n}}_i\widehat{\mathbf{n}}_i^T)\right]^{-1}\mathbf{s}}{|\mathbf{s}|} = (\mathbf{I} - 2\widehat{\mathbf{n}_2}\widehat{\mathbf{n}_2}^T)(\mathbf{I} - 2\widehat{\mathbf{n}_1}\widehat{\mathbf{n}_1}^T)\widehat{\mathbf{s}} \cdot \tag{29}$$

Finally, the TR of CR can be obtained by substituting Equations (27)–(29) into (14), i.e.,

$$\begin{aligned} \mathbf{E}_{\text{triple}}(\mathbf{r}) = jk\frac{e^{-jk(|\mathbf{r}|+|\mathbf{s}|)+j2\pi f_0 t}}{2\pi|\mathbf{r}||\mathbf{s}|}I_{\text{triple}} \times e^{2jk\widehat{\mathbf{s}}\cdot\sum_{m=1}^{2}\prod_{i=1}^{m}(\mathbf{I}-2\widehat{\mathbf{n}}_i\widehat{\mathbf{n}}_i^T)\widehat{\mathbf{n}}_m\widehat{\mathbf{n}}_m^T\mathbf{r}_{Am}} \\ \times\{(\widehat{\mathbf{n}_3}\cdot\mathbf{E}_2)[(\widehat{\mathbf{r}}\cdot\widehat{\mathbf{s}}_2)\widehat{\mathbf{r}} - \widehat{\mathbf{s}}_2] - (\mathbf{n}_3\cdot\widehat{\mathbf{s}}_2)[(\widehat{\mathbf{r}}\cdot\mathbf{E}_2)\widehat{\mathbf{r}} - \mathbf{E}_2]\} \end{aligned} \tag{30}$$

where $\widehat{\mathbf{n}_3}$ is the unit normal vector of TR's triangular facet, and $I_{\text{triple}} = \int\int_{s'} e^{jk(\widehat{\mathbf{r}}-\widehat{\mathbf{s}}_2)\cdot\mathbf{r}'}ds'$.

## 3. The CR Decomposition Based on the Main-Polarization

The SR, DR, and TR model of CR are shown in Equations (16), (26) and (30), respectively. When the incident wave's polarization changes, both the phase and amplitude

do not change. Thus, by ignoring the phase and amplitude, the polarizations of SR, DR, and TR are only related to

$$
\begin{aligned}
\mathbf{E}_{\text{single}}(\mathbf{r}) &\Rightarrow -\mathbf{E}_0 \\
\mathbf{E}_{\text{double}}(\mathbf{r}) &\Rightarrow (\widehat{\mathbf{n}_2} \cdot \mathbf{E}_1)[(\widehat{\mathbf{r}} \cdot \widehat{\mathbf{s}}_1)\widehat{\mathbf{r}} - \widehat{\mathbf{s}}_1] - (\widehat{\mathbf{n}_2} \cdot \widehat{\mathbf{s}}_1)[(\widehat{\mathbf{r}} \cdot \mathbf{E}_1)\widehat{\mathbf{r}} - \mathbf{E}_1] \\
\mathbf{E}_{\text{triple}}(\mathbf{r}) &\Rightarrow (\widehat{\mathbf{n}_3} \cdot \mathbf{E}_2)[(\widehat{\mathbf{r}} \cdot \widehat{\mathbf{s}}_2)\widehat{\mathbf{r}} - \widehat{\mathbf{s}}_2] - (\widehat{\mathbf{n}_3} \cdot \widehat{\mathbf{s}}_2)[(\widehat{\mathbf{r}} \cdot \mathbf{E}_2)\widehat{\mathbf{r}} - \mathbf{E}_2]
\end{aligned} \tag{31}
$$

First, let the horizontal polarization be $E_\phi$ and the vertical polarization be $E_\theta$. Taking SR in Equation (31) as an example, the reflected wave is $-E_\phi$ when the incident wave is $E_\phi$, and the reflected wave is $-E_\theta$ when the incident wave is $E_\theta$. Therefore, the PSM of SR is $[-1\ 0;\ 0\ -1]$. Similarly, the PSMs of SR, DR, and TR are

$$
\begin{cases}
\mathbf{E}_{\text{single}} = \begin{bmatrix} -1 & 0 \\ 0 & -1 \end{bmatrix} \\[2mm]
\mathbf{E}_{\text{double}} = \mathbf{U}_{p_i p_j} \begin{bmatrix} -1 & 0 \\ \chi_{p_i p_j} & 1 \end{bmatrix} \mathbf{U}_{p_i p_j}^{-1} \quad , \\[2mm]
\mathbf{E}_{\text{triple}} = \begin{bmatrix} 1 & 0 \\ 0 & 1 \end{bmatrix}
\end{cases} \tag{32}
$$

where $\chi_{p_i p_i}$ is determined by the ratio of DR from plane $p_i$ to plane $p_j$ and from plane $p_j$ to plane $p_i$, $\mathbf{U}_{p_i p_j}$ is

$$
\mathbf{U}_{p_i p_j} = \begin{bmatrix} \cos \psi_{p_i p_j} & -\sin \psi_{p_i p_j} \\ \sin \psi_{p_i p_j} & \cos \psi_{p_i p_j} \end{bmatrix} , \tag{33}
$$

where $\psi_{p_i p_j}$ is the rotation angle for DR from plane $p_i$ to plane $p_j$.

Then, we denote the odd scattering as the sum of SR and TR, and symbolize the even scattering as the DR. References [27–29] obtained the odd scattering and even scattering by the Krogager decomposition [49], which decomposes the scattering matrix into spherical scattering (odd scattering), dihedral scattering (even scattering), and helix scattering, i.e.,

$$
\mathbf{S} = e^{j\phi_0}\left\{ e^{j\phi_S} k_S \begin{bmatrix} 1 & 0 \\ 0 & 1 \end{bmatrix} + k_D \begin{bmatrix} \cos 2\theta_D & \sin 2\theta_D \\ \sin 2\theta_D & -\cos 2\theta_D \end{bmatrix} + k_H e^{\mp j2\theta_D} \begin{bmatrix} 1 & \mp j \\ \mp j & -1 \end{bmatrix} \right\} , \tag{34}
$$

where $k_S$, $k_D$ and $k_H$ are the coefficients of spherical scattering, dihedral scattering, and helix scattering, respectively. $\phi_0$ is the absolute phase of scattering matrix, $\phi_S$ is the relative phase of spherical scattering to dihedral scattering, and $\theta_D$ is the rotation of the target relative to the horizontal polarization.

Comparing Equations (32) and (34), we find that the Krogager decomposition has the following assumptions when performing the odd scattering and even scattering of CR.

1.　It assumes that the helix scattering is a noise after removing the odd scattering and even scattering, whereas the helix scattering is poorly representative of clutter;
2.　It assumes that the relative phase of dihedral scattering and helix scattering is 0;
3.　It assumes that the dihedral scattering only has a single dihedral scattering. Since there are three kinds of dihedral scatterings in trihedral CR, the dihedral scattering in Equation (34) cannot characterize the even scattering of CR.

In brief, the coefficient errors of odd scattering and even scattering after Krogager decomposition would increase as SCR decreases. To resolve the aforementioned problem, we propose a novel CR decomposition by the main-polarization as follows.

First, we denote the CR and clutter scattering matrix as $\mathbf{S}_C$ and $\mathbf{S}_N$, respectively. The scattering matrix of echo is

$$\mathbf{S} = \mathbf{S}_C + \mathbf{S}_N = \begin{bmatrix} S_{HH} & S_{HV} \\ S_{VH} & S_{VV} \end{bmatrix}, \tag{35}$$

where $S_{HV}$ means the horizontal transmitting and vertical receiving polarization, and the others are defined similarly.

For the scattering matrix $\mathbf{S}$ of $i$-th pulse, we transform $\mathbf{S}$ to Kennaugh matrix [50] as shown in Equation (36).

$$\mathbf{K}_i = \begin{bmatrix} A_0 + B_0 & C_k & H_k & F_k \\ C_k & A_0 + B_k & E_k & G_k \\ H_k & E_k & A_0 + B_k & D_k \\ F_k & G_k & D_k & A_0 + B_0 \end{bmatrix}, \tag{36}$$

where $A_0$, $B_0$, $B_k$, $C_k$, $D_k$, $E_k$, $F_k$, $G_k$ are the Huynen parameters [51].

Then, the average Kennaugh matrix during $N$ pulses is

$$\langle \mathbf{K} \rangle = \frac{1}{N} \sum_{i=1}^{N} \mathbf{K}_i. \tag{37}$$

Subsequently, the average Kennaugh matrix $\langle \mathbf{K} \rangle$ can be decomposed by Yang decomposition [52], i.e.,

$$\langle \mathbf{K} \rangle = \mathbf{K}_0 + \mathbf{K}_n, \tag{38}$$

where $\mathbf{K}_0$ is the Kennaugh matrix of CR, and $\mathbf{K}_n$ is the Kennaugh matrix of clutter after pulse accumulation.

Then, $\mathbf{K}_0$ is converted into a coherent matrix $\mathbf{T}_0$, viz.,

$$\mathbf{T}_0 = \begin{bmatrix} 2A_0 + B_0 & C_k - jD_k & H_k + jG_k \\ C_k + jD_k & B_0 + B_k & E_k + jF_k \\ H_k - jG_k & E_k - jF_k & B_0 - B_k \end{bmatrix}. \tag{39}$$

Then, the Pauli vector $\mathbf{k}_p = [k_1 \ k_2 \ k_3]^T$ is given using the coherence matrix $\mathbf{T}_0$, as shown in Equation (40).

$$\mathbf{T}_0 = \begin{bmatrix} |k_1|^2 & k_1 k_2^* & k_1 k_3^* \\ k_2 k_1^* & |k_2|^2 & k_2 k_3^* \\ k_3 k_1^* & k_3 k_2^* & |k_3|^2 \end{bmatrix}, \tag{40}$$

where $k_i^*$ is the conjugate value of $k_i$.

Therefore, the scattering matrix $\mathbf{S}_k$ of CR after denoising can be obtained, i.e.,

$$\mathbf{S}_k = k_1 \mathbf{S}_{p_1} + k_2 \mathbf{S}_{p_2} + k_3 \mathbf{S}_{p_3}, \tag{41}$$

where $\mathbf{S}_{p_1}$, $\mathbf{S}_{p_2}$, and $\mathbf{S}_{p_3}$ are the fundamental scatterings of Pauli decomposition [50].

In the above process, $\mathbf{S}_k$ is obtained using the Pauli vector $\mathbf{k}_p = [k_1 \ k_2 \ k_3]^T$, in which the coefficients are from the coherence matrix $\mathbf{T}_0$. By comparing the original scattering matrix $\mathbf{S}_C$, $\mathbf{S}_k$ only has an absolute phase difference, which does not affect the coefficients of the fundamental scatterings.

The trihedral CR mainly includes four basic scatterings, i.e., SR, DR, TR, and edge diffraction. When the radar incident wave irradiates the central axis of CR, TR is dominant.

However, SR, DR, and edge diffraction are dominant when the radar incident wave irradiates the edge or vertical plane of CR. Since the edge diffraction is still 20 dB lower than SR and DR [53,54], the decomposition of CR is exhibited in Equation (42) after ignoring the influence of edge diffraction.

$$\mathbf{S}_k = (\mathbf{S}_{SR} + \mathbf{S}_{TR}) + \mathbf{S}_{DR} = \mathbf{S}_{\text{odd}} + \mathbf{S}_{\text{even}} \,, \tag{42}$$

where $\mathbf{S}_{SR}$, $\mathbf{S}_{DR}$, $\mathbf{S}_{TR}$ are the SR, DR, and TR, respectively. $\mathbf{S}_{\text{odd}}$, $\mathbf{S}_{\text{even}}$ are odd scattering and even scattering, respectively.

For the CR odd scattering, it consists of SR and TR. Based on Equation (32), the odd scattering is

$$\begin{aligned}
\mathbf{S}_{\text{odd}} &= \mathbf{S}_{SR} + \mathbf{S}_{TR} \\
&= k_{SR}e^{j\theta_{SR}}\begin{bmatrix} 1 & 0 \\ 0 & 1 \end{bmatrix} + k_{TR}e^{j\theta_{TR}}\begin{bmatrix} 1 & 0 \\ 0 & 1 \end{bmatrix}, \\
&= k_{\text{odd}}e^{j\theta_{\text{odd}}}\begin{bmatrix} 1 & 0 \\ 0 & 1 \end{bmatrix}
\end{aligned} \tag{43}$$

where $k_{SR}$, $k_{TR}$ are the coefficients of SR and TR, respectively. $\theta_{SR}$, $\theta_{TR}$ are the phases of SR and TR, respectively. $k_{\text{odd}}$ and $\theta_{\text{odd}}$ are the coefficient and phase of odd scattering, respectively.

For the CR even scattering, it consists of three kinds of dihedral scatterings. Based on Equation (32), the even scattering is

$$\begin{aligned}
\mathbf{S}_{\text{even}} &= \mathbf{S}_{DR12} + \mathbf{S}_{DR23} + \mathbf{S}_{DR31} \\
&= k_{DR12}e^{j\theta_{DR12}}\mathbf{U}_{12}\begin{bmatrix} -1 & 0 \\ \chi_{12} & 1 \end{bmatrix}\mathbf{U}_{12}{}^{-1} + k_{DR23}e^{j\theta_{DR23}}\mathbf{U}_{23}\begin{bmatrix} -1 & 0 \\ \chi_{23} & 1 \end{bmatrix}\mathbf{U}_{23}{}^{-1} \\
&\quad + k_{DR31}e^{j\theta_{DR31}}\mathbf{U}_{31}\begin{bmatrix} -1 & 0 \\ \chi_{31} & 1 \end{bmatrix}\mathbf{U}_{31}{}^{-1} \\
&= k_{\text{even}}e^{j\theta_{\text{even}}}\begin{bmatrix} 1 & k_{\text{cross}_1}e^{j\Delta\theta_1} \\ k_{\text{cross}_2}e^{j\Delta\theta_2} & -1 \end{bmatrix}
\end{aligned} \tag{44}$$

where $\mathbf{S}_{DRp_ip_j}$ is the DR of planes $p_i$ and $p_j$, $\{p_i, p_j = 1,2,3\}$. $k_{DRp_ip_j}$, $\theta_{DRp_ip_j}$ are the coefficient and phase of $\mathbf{S}_{DRp_ip_j}$, respectively. $k_{\text{even}}$, $\theta_{\text{even}}$ are the even scattering's coefficient and phase, respectively. $k_{\text{cross}_i}$, $\Delta\theta_i$ are the cross-polarization's coefficient and phase, respectively.

After analyzing the basic scatterings of CR, the CR DR is formed by three kinds of dihedral scattering as shown in Equation (44). By comparing Equations (34) and (44), we find that the dihedral scattering in Krogager decomposition cannot accurately represent the CR DR. To resolve the above problem, we optimize the decomposition of CR by utilizing Equations (43) and (44), and then propose the CR decomposition method based on main polarization as follows.

By combining Equations (43) and (44), the scattering matrix $\mathbf{S}_k$ of CR after denoising is

$$\mathbf{S}_k = \mathbf{S}_{\text{odd}} + \mathbf{S}_{\text{even}} = k_{\text{odd}}e^{j\theta_{\text{odd}}}\begin{bmatrix} 1 & 0 \\ 0 & 1 \end{bmatrix} + k_{\text{even}}e^{j\theta_{\text{even}}}\begin{bmatrix} 1 & k_{\text{cross}_1}e^{j\Delta\theta_1} \\ k_{\text{cross}_2}e^{j\Delta\theta_2} & -1 \end{bmatrix}. \tag{45}$$

Subsequently, we utilize the main polarization of $\mathbf{S}_k$ to analyze the odd scattering and even scattering, i.e.,

$$\begin{aligned}
k_{\text{odd}}e^{j\theta_{\text{odd}}} + k_{\text{even}}e^{j\theta_{\text{even}}} &= \mathbf{S}_k[1,1] \\
k_{\text{odd}}e^{j\theta_{\text{odd}}} - k_{\text{even}}e^{j\theta_{\text{even}}} &= \mathbf{S}_k[2,2]
\end{aligned}, \tag{46}$$

where $\mathbf{S}_k[x, y]$ is the element of the $x$-th row and the $y$-th column of the matrix $\mathbf{S}_k$.

Based on Equations (36)–(41), (45), and (46), the coefficients $k_{\text{odd}}$ and $k_{\text{even}}$ of CR odd scattering and even scattering can be quickly obtained, thus completing the CR decomposition. The algorithm flow chart of CR decomposition is shown in Figure 4. First, we transform the Sinclair matrix into a Kennaugh matrix, and perform denoising by utilizing pulse accumulation and Yang decomposition. Then, we transform $\mathbf{K}_0$ to the Sinclair matrix by the Pauli vector. Finally, the coefficients of odd scattering and even scattering can be obtained by the main-polarization decomposition. With the CR decomposition, we cannot only cure the shortcomings of Krogager decomposition, but also accurately obtain the odd scattering and even scattering of CR in clutter.

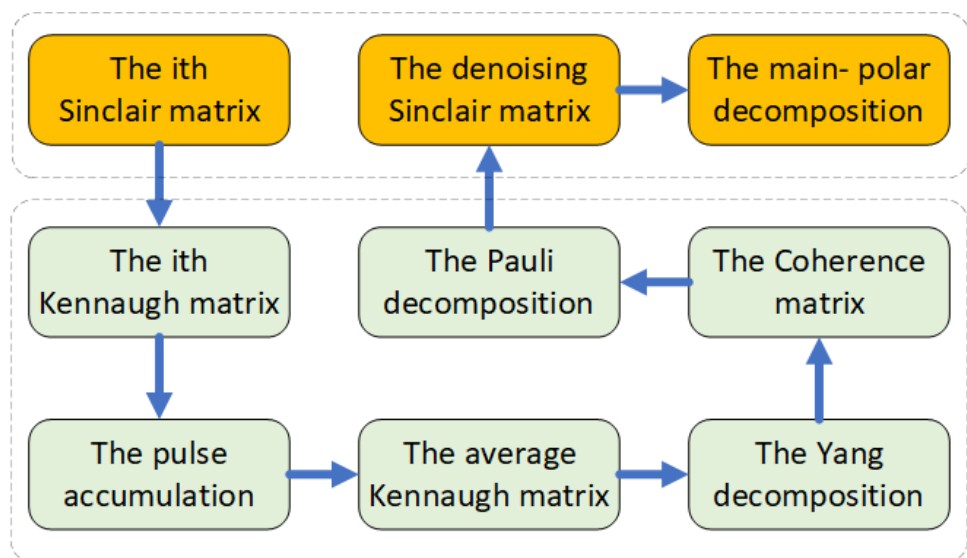

**Figure 4.** The CR decomposition.

## 4. The CR Discrimination Method Based on TSP Joint Domains

A new CR decomposition method is proposed in Section 3, and we can obtain the polarization characteristic of CR according to the above new CR decomposition method. On this basis, we integrate the spatial–time characteristic of the radar and target into the polarization characteristic, thereby proposing a discrimination method of CR, the details of which are as follows.

### 4.1. The Polarization Characteristic

With the coefficients of the CR odd scattering and even scattering, this section will analyze the polarization characteristic of CR, thus facilitating to the distinction between CR and ship. First, we introduce the definition of MSP as follows.

**Definition 1.** *The CR mainly includes SR, DR, and TR. The SR and TR belong to the odd scattering, and DR belongs to the even scattering. At a certain azimuth and pitch angle, the even scattering is the main scattering polarization (MSP) if the even scattering is stronger than odd scattering. On the contrary, the odd scattering is MSP.*

By analyzing the basic scatterings of odd scattering and even scattering in trihedral CR, we find that the SR, DR, and TR are all single-peak functions, the analysis and simulation of which are shown in Appendix A. Combining the changing rule of odd scattering and even scattering, the polarization characteristic is given in Appendix A, i.e., the MSP coefficient of CR is a single-peak curve, when the azimuth or pitch angle of radar incident wave changes monotonically within $35°$.

### 4.2. The Spatial–Time Characteristic

Section 4.1 gives the polarization characteristic of CR, which is a characteristic which has come from the mutual motion of the radar and target. For a moment of stillness, the aforementioned polarization characteristic does not exist. Therefore, we need to consider the mutual movement of the radar and target, and then ensure the existence condition of the polarization characteristic in Section 4.1. In the spatial–time characteristic of the radar and target, the radar usually detects a ship on the sea surface from the air, and finally attacks the ship in the form of diving to increase the effect and speed. Actually, a large pitch angle attack is required when hitting the target, i.e., striking a ship from a nearly vertical direction, thus improving the strike capability. This section will take the large pitch angle attack as an example to analyze the spatial-time characteristic of the radar and target, which is based on the three-dimensional proportional guidance method.

The spatial relationship of the radar and target is shown in Figure 5. The radar's position is $O$, the target is $T_0$, and the LOS between the radar and target is $\overrightarrow{OT_0}$. $OX_gY_gZ_g$, $OX_tY_tZ_t$ is the radar and target's inertial coordinate system, respectively. $\varphi_L$, $\theta_L$ are the LOS azimuth and pitch angle, respectively. $OX_lY_lZ_l$ is the LOS coordinate system, where $\overrightarrow{OX_l}$ is consistent with the LOS direction, $\overrightarrow{OZ_l}$ is downward in the vertical plane and also orthogonal to $\overrightarrow{OX_l}$, $\overrightarrow{OY_l}$ is determined based on the right-hand rule. The velocity of radar is $\mathbf{V}_O$, and $\varphi_V$, $\theta_V$ are the velocity azimuth and pitch angle, respectively. $OX_vY_vZ_v$ is the radar velocity's coordinate system, where $\overrightarrow{OX_v}$ is consistent with the velocity direction, $\overrightarrow{OZ_v}$ is downward in the vertical plane as well as orthogonal to $\overrightarrow{OX_v}$, and $\overrightarrow{OY_v}$ is determined by the right-hand rule. The velocity of the target is $\mathbf{V}_t$, and its azimuth and pitch angle are $\varphi_T$ and $\theta_T$, respectively.

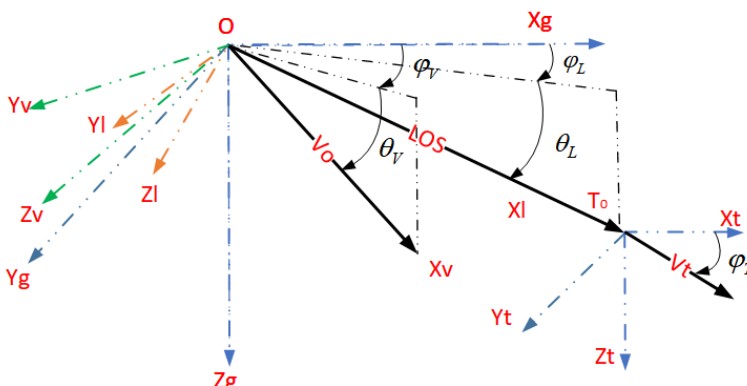

**Figure 5.** The spatial relationship between radar and target.

There are three coordinate systems in Figure 5, i.e., the inertial coordinate system $OX_gY_gZ_g$, the LOS coordinate system $OX_lY_lZ_l$, and the velocity coordinate system $OX_vY_vZ_v$. The rotation matrices between the three coordinate systems are exhibited in Equation (47), which can be used to transform $OX_gY_gZ_g$ into $OX_lY_lZ_l$ or $OX_vY_vZ_v$.

$$
\begin{aligned}
\mathbf{C}_{g\to l} &= \begin{bmatrix} \cos\theta_L\cos\varphi_L & \cos\theta_L\sin\varphi_L & \sin\theta_L \\ -\sin\varphi_L & \cos\varphi_L & 0 \\ -\sin\theta_L\cos\varphi_L & -\sin\theta_L\sin\varphi_L & \cos\theta_L \end{bmatrix} \\
\mathbf{C}_{g\to v} &= \begin{bmatrix} \cos\theta_V\cos\varphi_V & \cos\theta_V\sin\varphi_V & \sin\theta_V \\ -\sin\varphi_V & \cos\varphi_V & 0 \\ -\sin\theta_V\cos\varphi_V & -\sin\theta_V\sin\varphi_V & \cos\theta_V \end{bmatrix}
\end{aligned}.
\tag{47}
$$

For a brief description, we take the pitch plane as an example in the following. First, an offset term of angular rate is

$$w_\theta = \frac{\eta_\theta(\theta_d - \theta_L)}{N_\theta t} = \frac{\eta_\theta v_x^{LOS}(\theta_d - \theta_L)}{N_\theta r},$$ (48)

where $\eta_\theta$ and $N_\theta$ both are guidance constants, $\theta_d$ is the constrained pitch angle when striking target, and $t$ is the remaining time.

Subsequently, the LOS velocity $v_x^{LOS}$ can be obtained as shown in Equation (49), i.e., by transferring the radar velocity from the velocity coordinate system to an LOS coordinate system.

$$\begin{bmatrix} v_x^{LOS} \\ v_y^{LOS} \\ v_z^{LOS} \end{bmatrix} = \mathbf{C}_{g\to l}\mathbf{C}_{g\to v}^T \begin{bmatrix} v_x^o \\ v_y^o \\ v_z^o \end{bmatrix},$$ (49)

where $v_x^o, v_y^o, v_z^o$ are the components of radar velocity in a velocity coordinate system, respectively.

Therefore, the injunction of a pitch rate in the proportional guidance is

$$\Omega_\theta = N_\theta \left( w_\theta{}^{LOS} - w_\theta \right),$$ (50)

where $w_\theta{}^{LOS}$ is the actual change rate of the LOS pitch angle.

In the same way, the injunction of an azimuth rate in the proportional guidance is

$$\Omega_\varphi = N_\varphi \left( w_\varphi{}^{LOS} - \frac{\eta_\varphi v_x^{LOS}(\varphi_d - \varphi_L)}{N_\varphi r} \right),$$ (51)

where $\eta_\varphi$ and $N_\varphi$ both are navigation constants, $\varphi_d$ is the constrained azimuth angle when striking the target, and $w_\varphi{}^{LOS}$ is the actual change rate of the LOS azimuth angle.

In brief, the accelerated velocity of a radar can be controlled by iteratively updating $\Omega_\theta$ and $\Omega_\varphi$ with Equations (50) and (51). Therefore, we can successfully perform the three-dimensional proportional guidance with angle constraints, which can control the radar-to-strike target at a constrained angle. Now, we take an example to briefly explain the spatial–time characteristic in a terminal guidance process.

**Example 1.** *The LOS distance between radar and ship is* 20 *km, the initial LOS azimuth angle* $\varphi_L$ *is* 30°*, and the initial LOS pitch angle* $\theta_L$ *is* 45°*. The radar velocity* $\mathbf{V}_O$ *is* 1000 *m/s, the radar velocity's azimuth angle* $\varphi_V$ *is* −10°*, and the radar velocity's pitch angle* $\theta_V$ *is* 40°*. The ship velocity* $\mathbf{V}_t$ *is* 15 *m/s, the ship velocity's azimuth angle* $\varphi_T$ *is* 20°*, and the ship velocity's pitch angle* $\dot\theta_T$ *is* 0°*.*

In the three-dimensional proportional guidance, the pitch constraint angle $\theta_d$ is 90°, and the azimuth constraint angle $\varphi_d$ is not limited. The trajectory of the radar is displayed in Figure 6, and the changes in the LOS azimuth angle and pitch angle are shown in Figure 7. We find that both the azimuth angle and pitch angle are monotonic changing functions. To sufficiently demonstrate the above conclusion, we randomly adjusted the initial LOS azimuth angle $\varphi_L$, the initial LOS pitch angle $\theta_L$, the radar velocity's azimuth angle $\varphi_V$, the radar velocity's pitch angle $\theta_V$, and the ship velocity's azimuth angle $\varphi_T$. After 10,000 Monte Carlo simulations, we also found that the LOS azimuth angle and pitch angle both change monotonically when the radar approaches the target, which is the spatial–time characteristic of the radar in the terminal guidance process.

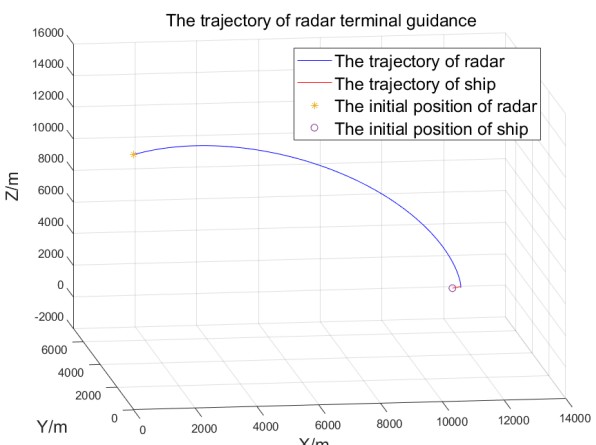

**Figure 6.** The trajectory of radar terminal guidance.

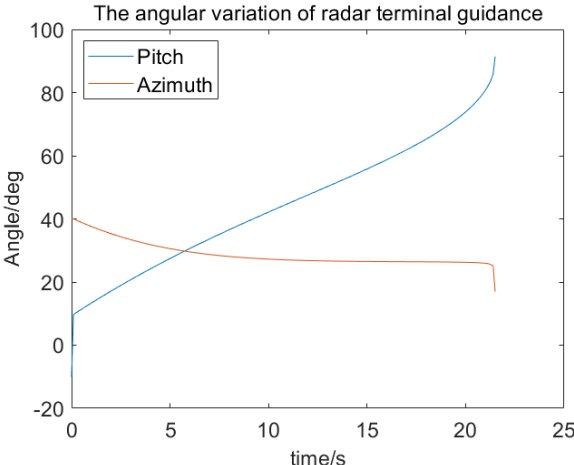

**Figure 7.** The angular variation of radar terminal guidance.

### 4.3. The Discrimination Method

With the analyses of the polarization characteristic and the spatial–time characteristic, we find that the MSP of CR is a single-peak curve when radar approaches the ship. In contrast, the ship's odd scattering and even scattering will change irregularly as the LOS changes, and because of this, the ship's strong scattering points are unstable. Accordingly, the MSP of a ship is not a single-peak curve, which is different to that of CR. In order to quantitively describe the MSP amplitude curve, this paper introduces a CR-MSE parameter as follows.

First, we carry out the MSP amplitude within a period of time, and then utilize the quadratic function for curve fitting. The MSE between the MSP and quadratic function is

$$S_{\mathrm{MSE}} = \frac{1}{N} \sum_{i=1}^{N} (P_i - f_i)^2, \tag{52}$$

where $N$ is the number of target echoes after pulse accumulation, $P_i$ is the MSP amplitude, and $f_i$ is the quadratic function of MSP.

Due to fact that the MSP of CR is a single-peak curve and that the MSP of a ship changes irregularly, the $S_{\mathrm{MSE}}$ of CR is much smaller than that of a ship. Therefore, $S_{\mathrm{MSE}}$ can be utilized as a parameter to discriminate CRs and ships, i.e.,

$$\mathrm{Target} = \begin{cases} \mathrm{Corner\ \ Reflector} \ , & S_{\mathrm{MSE}} < T_{\mathrm{value}} \\ \mathrm{Ship} \ \ \ \ \ \ \ \ \ , & S_{\mathrm{MSE}} \geq T_{\mathrm{value}} \end{cases}, \tag{53}$$

where $T_{value}$ is the judgment threshold. Since the judgment threshold is related to the polarization characteristic of CR, it is difficult to give a numerical expression from the perspective of probability distribution. Therefore, this paper analyzes the judgment thresholds under different discrimination probabilities and different false alarm probabilities by means of a large number of simulations. The details are given in Section 5.3.

Taking the influence of clutter into account, we propose a CR discrimination method, which is composed of the CR decomposition and Equations (52) and (53). The algorithm flow chart is shown in Figure 8, and the detailed steps are as follows.

Step 1: For the scattering matrices of $M$ pulses in a period of time, we perform pulse accumulation every $N$ pulses;

Step 2: The $M/N$ MSPs are given by the CR decomposition using Equations (36)–(41), (45), and (46);

Step 3: The MSP is fitted by the quadratic function, so as to calculate $S_{MSE}$ using Equation (52);

Step 4: The ship and CR can be discriminated by their $S_{MSE}$ using Equation (53), where it is judged as a CR if $S_{MSE} < T_{value}$, otherwise it is a ship.

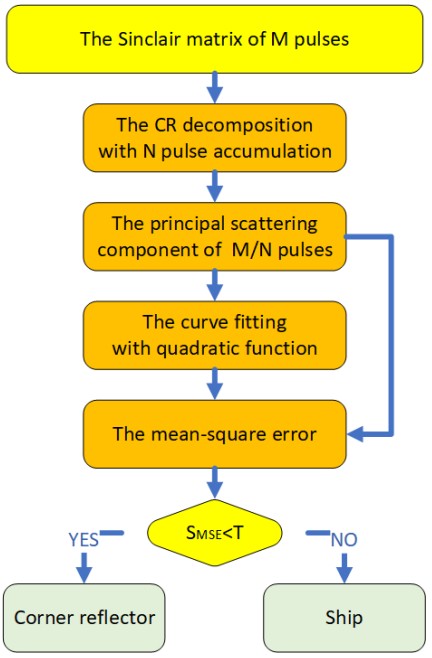

**Figure 8.** The CR discrimination method based on TSP joint domains.

## 5. Simulation

### 5.1. The CR Polarization Characteristic

In Section 4, we analyzed that the CR MSP is a single-peak curve when the azimuth or pitch angle changes monotonically within 35° without clutter. Therefore, this section will demonstrate the above conclusion by analyzing the MSP amplitude under any angle change.

**Example 2.** *The length, width, and height of CR are all 0.25 m. The frequency of incident wave is 12 GHz. Therefore, the wavelength $\lambda$ is 0.025 m, and the dimension of CR satisfies $10\lambda$. Subsequently, the surface of CR is divided by a triangular surface, and the length of triangular surface is less than $\lambda/10$. The range of azimuth and pitch angle are both $[0°, 90°]$, and the angular interval is $1°$. With the change in azimuth and pitch angle, the coefficients of odd scattering and even scattering are calculated.*

First, we randomly choose the initial pitch angle $\theta$ and azimuth angle $\varphi$, i.e., $\theta = 50°$ and $\varphi = 20°$. The monotonic change curves of the pitch angle and azimuth angle are

randomly given and are satisfied within 35°, as exhibited in Figure 9. Subsequently, we calculate the CR amplitude changes of different scatterings, i.e., the odd scattering, even scattering, and MSP, as shown in Figure 10. However, the MSP is obtained by Equation (54).

$$\begin{cases} P_{\text{MSP}}(i) = \text{odd}(i)\,, & if \ \ \text{odd}(i) > \text{even}(i) \\ P_{\text{MSP}}(i) = \text{even}(i), & if \ \ \text{odd}(i) \le \text{even}(i) \end{cases}, \tag{54}$$

where $P_{\text{MSP}}(i)$ is MSP. $\text{odd}(i)$, $\text{even}(i)$ are the coefficients of odd scattering and even scattering, respectively.

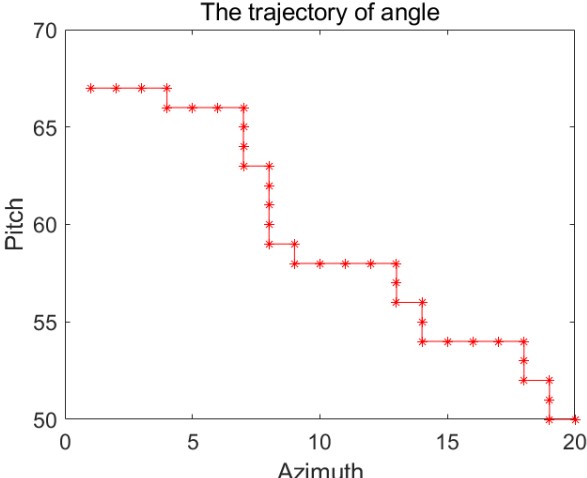

**Figure 9.** The changes of pitch and azimuth angle.

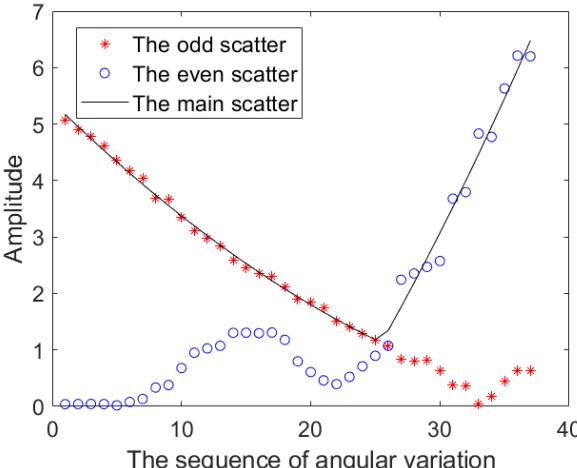

**Figure 10.** The amplitude changes of odd scattering, even scattering, and MSP.

Then, we analyze the number of MSP changes by odd scattering or even scattering, viz.,

$$P_{\text{change}} = \sum_{i=2}^{M} \eta(i) \begin{cases} \eta(i) = 1, & if \ k(i-1)k(i) < 0 \\ \eta(i) = 0, & else \end{cases}, \tag{55}$$

where $M$ is the number of pulses required for discrimination and $k(i)$ represents whether MSP is composed of odd scattering or even scattering, as shown in Equation (56).

$$\begin{cases} k(i) = 1, & if \ \ \text{odd}(i) > \text{even}(i) \\ k(i) = -1, & if \ \ \text{odd}(i) \le \text{even}(i) \end{cases}. \tag{56}$$

Finally, the monotonic changing curves of the pitch angle and azimuth angle are randomly generated by utilizing 10,000 Monte Carlo simulations, which can satisfy the changing range within 35°. By Equations (54)–(56), the numbers of MSP changing $P_{\text{change}}$ can be obtained as displayed in Table 1. The results show that the changes of MSP are mainly 0 and 1. If $P_{\text{change}} = 0$, MSP is only odd scattering or even scattering. If $P_{\text{change}} = 1$, MSP is composed of "odd scattering + even scattering" or "even scattering + odd scattering". Because of this, the odd scattering and even scattering are both monotonic functions, and the changing trend of MSP is mainly a monotonic function (i.e., a special single-peak function) or single-peak function, thus demonstrating the polarization characteristic of CR.

**Table 1.** The number of MSP changes.

| $P_{\text{change}}$ | 0 | 1 | 2 |
|---|---|---|---|
| Number | 9607 | 379 | 14 |

### 5.2. The CR Discrimination Method

Focusing on the CR discrimination method based on TPS joint domains, this section will analyze the discrimination probability under different observing time and different SCRs.

**Example 3.** *The simulation model of CR is the same as that of Example 2. Now, the same triangular facet division and incident wave frequency are used to simulate the ship model. The ship's length is 116 m, width is 28 m, and height is 39 m, as shown in Figure 11. Subsequently, the trajectories of radar and target are simulated. The LOS distance between the radar and ship is 20 km, the radar velocity $|V_O|$ is 1000 m/s, the ship velocity $|V_t|$ is 15 m/s, and the direction parameters are presented in Table 2. Among them, the radar velocity's azimuth angle $\varphi_V$ is randomly selected within 30° more or less than the initial LOS azimuth angle $\varphi_L$. The radar strikes the ship with a three-dimensional proportional guidance method based on angle constraints, where the pitch constraint angle $\theta_d$ is 90°, and the azimuth constraint angle $\varphi_d$ is not limited.*

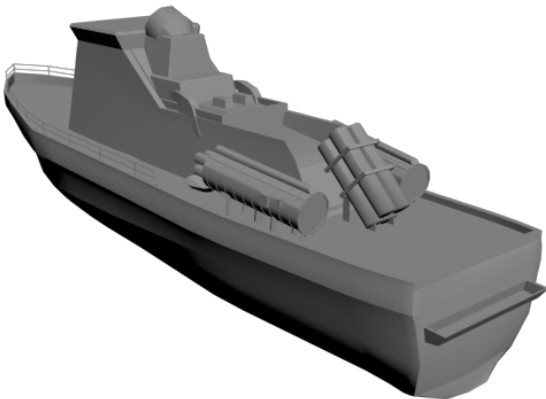

**Figure 11.** The model of a ship.

**Table 2.** The direction parameters of the radar and ship.

| Type | The Initial LOS | The Velocity of Radar | The Velocity of Ship |
|---|---|---|---|
| Azimuth | $\varphi_L = $ random | $\varphi_V = $ random | $\varphi_T = $ random |
| Pitch | $\theta_L = 45°$ | $\theta_V = $ random | $\theta_T = 0°$ |

First, let the observing time be 10 s, and randomly select the direction parameters. Then, we perform 10,000 Monte Carlo simulations to calculate the CR-MSEs of CR and ship according to Equation (52), as displayed in Figure 12. The result shows that the

average CR CR-MSE is 0.0003, whereas that of the ship is 0.0041. Therefore, the CR CR-MSE is much smaller than that of ship, which demonstrates that the CR and ship can be discriminated based on CR-MSE. By changing the judgment threshold with a large number of simulations, the discrimination probability and false alarm probability are exhibited in Figure 13. As the discrimination probability increases, the false alarm probability increases. Therefore, we can determine the judgment threshold based on Figure 13, when the discrimination probability and false alarm probability both are fixed. Regarding the discrimination of corner reflectors, all methods (i.e., HRRP, micro-Doppler, and polarization methods in the Introduction) need to obtain a threshold level using simulation or actual data. The method in this paper is the same as the above methods, which fails to give a closed-form formula for the threshold $T_{value}$. However, compared with other methods, the threshold in this paper is obtained by combining the spatial–time characteristic and polarization characteristic. Therefore, the adaptability of the threshold $T_{value}$ is more extensive. For example, for different scenarios with different parameters, i.e., the initial LOS, the velocity of the radar and the velocity of the ship, the discrimination probability, and false alarm probability of the present method are better than those of other methods, as shown in Example 5.

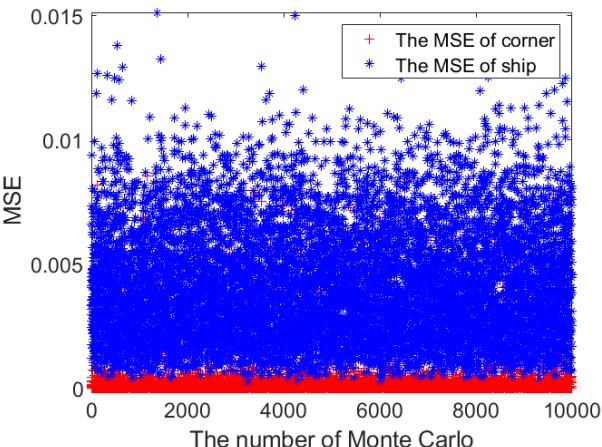

**Figure 12.** The CR-MSE of the CR and ship.

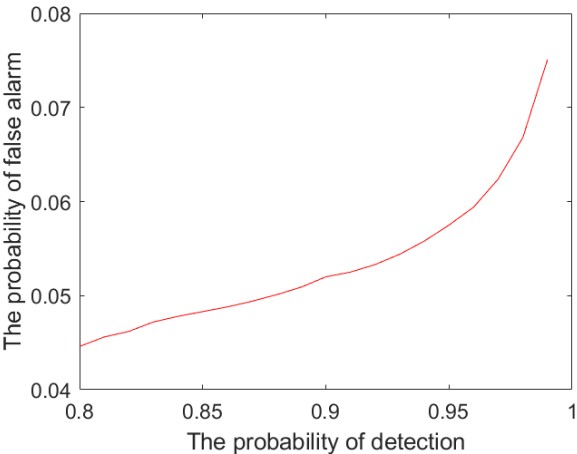

**Figure 13.** The discrimination probability and false alarm probability.

Subsequently, the observing time is chosen to be between 1–15 s, respectively. Based on the 10,000 Monte Carlo simulations, the judgment threshold is counted under the premise that the discrimination probability is 95%. Therefore, the false alarm probability can be calculated, as shown in Figure 14. As the observing time increases, the false alarm probability first decreases and then increases. The result can indicate that the fluctuation of

the ship's MSP is not obvious when the observing time is short. At this time, the judgment threshold is low, which will increase the false alarm probability of CR. Nevertheless, the changing trend of the CR MSP will fluctuate when the observing time is long, which also increases the false alarm probability of the CR. Therefore, the difference between the CR and ship is the most obvious when the observing time is 7 s under the premise of ensuring the discrimination probability of 95%. At this time, the false alarm probability is 4.1%.

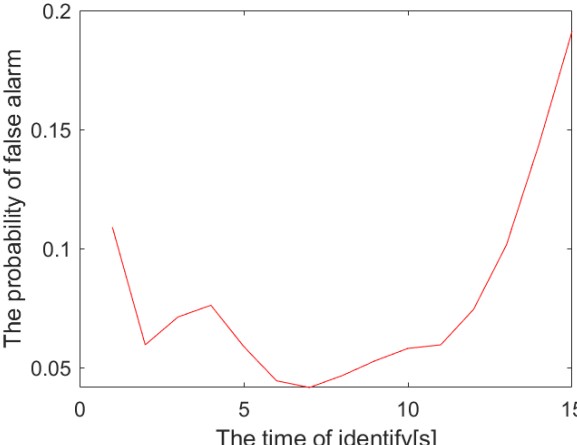

**Figure 14.** Probability of false alarm under different discrimination times.

After determining the observation time and threshold, we will analyze the discrimination probability and false alarm probability under different SCRs. Let the SCR change in the scope of [0 dB, 20 dB], whilst the discrimination probability and false alarm probability are shown in Figures 15 and 16. The discrimination probability and false alarm probability under different SCRs are mainly closely related to the denoising effect of CR decomposition. In Figure 17, we will find that the CR decomposition has a better denoising effect when the SCR is higher than 7 dB. Therefore, the discrimination probability is higher than 94% and the false alarm probability is lower than 7% when the SCR is higher than 7 dB, as exhibited in Figures 15 and 16. As the SCR increases, the discrimination probability will tend towards 95% and the false alarm probability will tend to 4.1%, which is consistent with the theoretical analysis of the CR decomposition, thus demonstrating that the present method is effective in clutter.

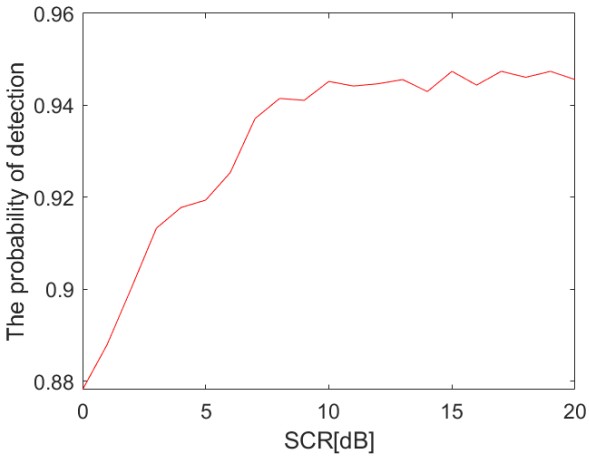

**Figure 15.** The discrimination probability under different SCRs.

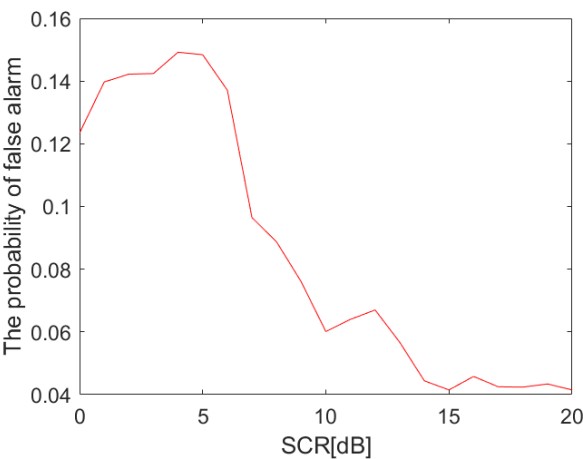

**Figure 16.** The false alarm probability under different SCRs.

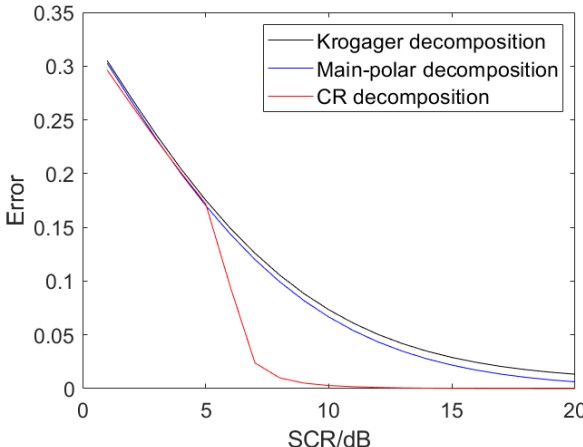

**Figure 17.** The errors of different decompositions.

*5.3. The Comparisons of the Proposed Work*

In this paper, we proposed a new polarization decomposition and a CR discrimination method. Therefore, we will give the comparisons of the proposed work in this section.

First, as for the new polarization decomposition, we will compare the coefficient errors of odd scattering and even scattering, which are based on the Krogager decomposition, main polarization decomposition, and the present method in this paper under different SCRs.

**Example 4.** *The simulation parameters are the same as that of Example 2. The range of the azimuth and pitch angle are both* $[0°, 90°]$, *and the angular interval is* $1°$. *The scattering matrices of SR, DR, and TR were obtained by the MoM in Feko software.*

After calculating the true values of odd scattering and even scattering, the clutter is added into four polarization channels, respectively. Subsequently, the coefficients of odd scattering and even scattering are performed by Krogager decomposition, main-polarization decomposition, and the present method, respectively. The coefficient errors are compared by Equation (57) as follows.

$$\text{error} = \frac{\sum\limits_{i=1}^{90} \sum\limits_{j=1}^{90} \left[ (C_{\text{odd}}(i,j) - D_{\text{odd}}(i,j))^2 + (C_{\text{even}}(i,j) - D_{\text{even}}(i,j))^2 \right]}{3600}, \qquad (57)$$

where $C_{\text{odd}}(i,j)$, $C_{\text{even}}(i,j)$ are the true values of odd scattering and even scattering under different azimuth and pitch angles, respectively. $D_{\text{odd}}(i,j)$, $D_{\text{even}}(i,j)$ are the odd scattering and even scattering coefficients by the decomposition method, respectively.

In order to analyze the influence of clutter, SCR will change in the scope of [0 dB, 20 dB]. Then, the error coefficients of Krogager decomposition, main-polarization decomposition, and the present method are displayed in Figure 17. By comparing the Krogager decomposition and main-polarization decomposition, the errors of two decomposition methods are not much different when SCR is small. However, the main-polarization decomposition has a smaller error than Krogager decomposition when the SCR is higher than 5 dB. This result shows that the dihedral scattering of Krogager decomposition cannot accurately represent the CR even scattering due it being constituted by three different DRs as shown in Equation (44). By observing the present method in Figure 17, its error is obviously better than that of Krogager decomposition and main-polarization decomposition when the SCR is higher than 7 dB, due to the CR decomposition utilizing the denoise function of the Yang decomposition. When SCR > 7 dB, the Yang decomposition can better remove noise, but its denoising effect is not obvious when the SCR is too low. Therefore, the compared results can demonstrate that the present method has a better denoising effect and is more adaptive to decompose the CR odd scattering and even scattering in clutter.

Secondly, we will compare the CR discrimination method in this paper with the HRRP method [14] and the polarization method [31].

**Example 5.** *The ship model, ship velocity, radar velocity, and relative range are the same as that of Example 3. The array CRs are composed by four CRs, for which the relative locations are [0 m, 29 m, 63 m, 117 m] and the relative RCS is [1, 0.8, 2, 1]. For the parameters of the radar, its frequency is 12 GHz, its bandwidth is 30 MHz, its pulse width is 2 μs, sampling frequency is 100 MHz, and SCR is 18 dB.*

In [14], the extreme learning machine algorithm is proposed to discriminate between ship and CR based on the HRRP differences. For example, the HRRPs of the ship and array CRs are displayed in Figures 18 and 19. In [31], six polarization parameters are utilized to discriminate between the ship and CR, i.e., the determinant value of PSM, trace of PSM, trace of power matrix, determinant value of power matrix, polarization shape factor, and combined invariant coefficient. In 10,000 Monte Carlo trials, we randomly selected the space and velocity parameters of the radar and target, i.e., the initial LOS, the velocity of the radar and the velocity of the ship. By performing 10,000 Monte Carlo simulations, the discrimination probability and false alarm probability of the three methods are exhibited in Table 3. For the HRRP method, the results are very susceptible to the arrangement between the CR and type of ship, thus having the highest false alarm probability. For the polarization method, it only utilizes the polarization information at an independent moment, which has an error when the incident wave is far from the LOS symmetry axis. In contrast, the method presented in this paper has comprehensively utilized the time, spatial, and polarization information. With the increases in the use of discriminative information and the expansion of the incident angle, the present method has higher discrimination probability and lower false alarm probability.

**Table 3.** The comparison of three methods.

| Probability | HRRP Method | Polarization Method | TSP Method |
|---|---|---|---|
| Discrimination | 91.5% | 92.3% | 94.7% |
| False alarm | 14.4% | 8.8% | 4.1% |

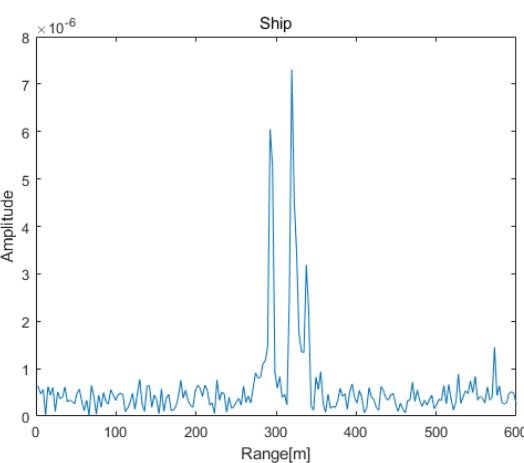

**Figure 18.** The HRRP of a ship.

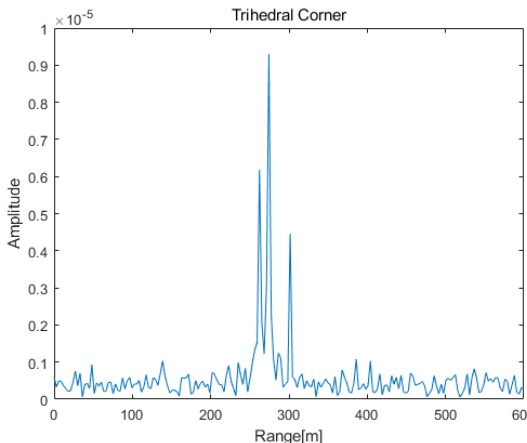

**Figure 19.** The HRRP of array CRs.

## 6. Conclusions

In this paper, we proposed a novel discrimination method based on high-frequency electromagnetic theory to discriminate the CR and ship. First, we proposed CR decomposition based on the main polarization, which could accurately obtain the coefficients of CR odd scattering and even scattering in clutter. Then, we gave the polarization characteristic of CR, i.e., the MSP amplitude of CR was a single-peak curve under ideal situation without clutter, when the azimuth or pitch angle of an incident wave changed monotonically within 35°. Subsequently, we analyzed the spatial–time characteristic based on the three-dimensional proportional guidance. Finally, we proposed a novel CR discrimination method based on the TSP joint domains by constructing the CR-MSE parameter, which could achieve 95% discrimination probability and 4.1% false alarm probability.

**Author Contributions:** Conceptualization, H.Y.; Investigation, H.H.; Methodology, Y.H. and J.Y. (Jian Yang); Writing—review & editing, J.Y. (Junjun Yin). All authors have read and agreed to the published version of the manuscript.

**Funding:** This research received no external funding.

**Data Availability Statement:** Not applicable.

**Conflicts of Interest:** The authors declare no conflict of interest.

## Appendix A. The Polarization Characteristic

For the monostatic radar detecting CR, the incident wave is $r$, the azimuth angle is $\varphi$, and the pitch angle is $\theta$, as shown in Figure A1. Let the planes $OAB$, $OBC$, and $OCA$ be

planes 1, 2, and 3, respectively. Then, $\mathbf{S}_{SR1}$, $\mathbf{S}_{SR2}$, and $\mathbf{S}_{SR3}$ are the SR of planes 1, 2, and 3, respectively. Among planes 1 and 2, planes 2 and 3, and planes 3 and 1, the DR in between are taken as $\mathbf{S}_{DR12}$, $\mathbf{S}_{DR23}$, and $\mathbf{S}_{DR31}$, respectively. The TR is $\mathbf{S}_{TR}$.

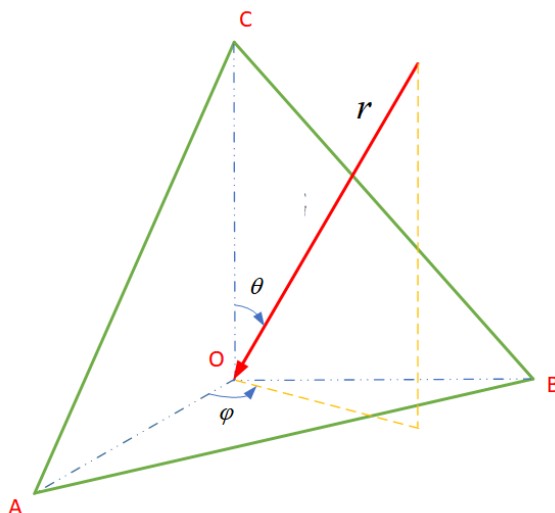

**Figure A1.** Schematic diagram of the incident wave.

First, the odd scattering matrix of CR is

$$\mathbf{S}_{\text{odd}} = \mathbf{S}_{SR1} + \mathbf{S}_{SR2} + \mathbf{S}_{SR3} + \mathbf{S}_{TR}. \tag{A1}$$

As for the $\mathbf{S}_{SR1}$ of plane 1, the coefficient of $S_{SR1}$ is the largest when the incident wave is perpendicular to plane 1. Moreover, $S_{SR1}$ is a monotonically decreasing function as the incident wave moves away from the normal line of the plane 1. In the same way, we find that both $\mathbf{S}_{SR2}$ and $\mathbf{S}_{SR3}$ are monotonically decreasing functions. Additionally, the maximum value of $\mathbf{S}_{SR2}$ is located on the normal line of plane 2, and the maximum value of $\mathbf{S}_{SR3}$ is located on the normal line of plane 3. As for $\mathbf{S}_{TR}$, the coefficient of $\mathbf{S}_{TR}$ is the largest when the incident wave is located on the LOS symmetry axis of CR (i.e., $\varphi = 45°$ and $\theta = 54°$). When the incident wave is far from the LOS symmetry axis, $\mathbf{S}_{TR}$ is also a monotonically decreasing function.

Subsequently, the even scattering matrix of CR is

$$\mathbf{S}_{\text{even}} = \mathbf{S}_{DR12} + \mathbf{S}_{DR23} + \mathbf{S}_{DR31}. \tag{A2}$$

For $\mathbf{S}_{DR12}$ between planes 1 and 2, the coefficient of $\mathbf{S}_{DR12}$ is the largest when the incident wave is on the symmetry plane of dihedral CR (i.e., $A - B - C - O - A$) and is also parallel to the bottom surface $OCA$. For the sake of brevity, we define the symmetrical parallel axis of dihedral CR, i.e., it is located on the symmetry plane of dihedral CR and is also parallel to the bottom surface at the same time. As the incident wave moves away from the symmetrical parallel axis of planes 1 and 2, $\mathbf{S}_{DR12}$ is a monotonically decreasing function. In the same way, $\mathbf{S}_{DR23}$ and $\mathbf{S}_{DR31}$ are both monotonically decreasing functions. Meanwhile, the maximum value of $\mathbf{S}_{DR23}$ is located on the symmetrical parallel axis of planes 2 and 3, and the maximum value of $\mathbf{S}_{DR31}$ is located on the symmetrical parallel axis of planes 3 and 1.

In summary, by analyzing the basic scatterings of odd scattering and even scattering in trihedral CR, we find that the SR (i.e., $S_{SR1}$, $S_{SR2}$, and $S_{SR3}$), DR (i.e., $S_{DR12}$, $S_{DR23}$, and $S_{DR31}$), and TR (i.e., $S_{TR}$) are all single-peak functions. To further analyze the changing rule of odd scattering and even scattering, we give a simulation in the following.

**Example A1.** *The length, width, and height of CR are all 0.25 m. The frequency of the incident wave is 12 GHz. Therefore, the wavelength $\lambda$ is 0.025 m, and the dimension of CR satisfies $10\lambda$. Subsequently, the surface of CR is divided by a triangular surface, and the length of a triangular surface is less than $\lambda/10$. We utilize the method of moments (MoM) in Feko software to simulate CR, and the range of azimuth and pitch angle both are $[0°, 90°]$ where the angular interval is $1°$.*

Aiming at the fully polarimetric data, the coefficients of CR odd scattering and even scattering are obtained by CR decomposition, as shown in Figures A2 and A3.

Figure A2 exhibits that how the coefficient of CR odd scattering changes in function of pitch and azimuth angles. The SR $\mathbf{S}_{SR1}$ is the largest in the normal line of plane 1, corresponding to the areas $\varphi = 0 - 90°$ and $\theta = 0°$ in Figure A2. The SR $\mathbf{S}_{SR2}$ is the largest in the normal line of plane 2, corresponding to the area $\varphi = 0°$ and $\theta = 90°$. The SR $\mathbf{S}_{SR3}$ is the largest in the normal line of plane 3, corresponding to the area $\varphi = 90°$ and $\theta = 90°$. The TR $\mathbf{S}_{TR}$ is the largest in the LOS symmetry axis of CR, corresponding to the area $\varphi = 45°$ and $\theta = 54°$. For the odd scattering containing $\mathbf{S}_{SR1}$, $\mathbf{S}_{SR2}$, $\mathbf{S}_{SR3}$, and $\mathbf{S}_{TR}$ at the same time, the odd scattering is a single-peak curve when the azimuth or pitch angle changes within $35°$, which is the minimum of $[35°, 45°, 90°]$, as displayed in Figure A2.

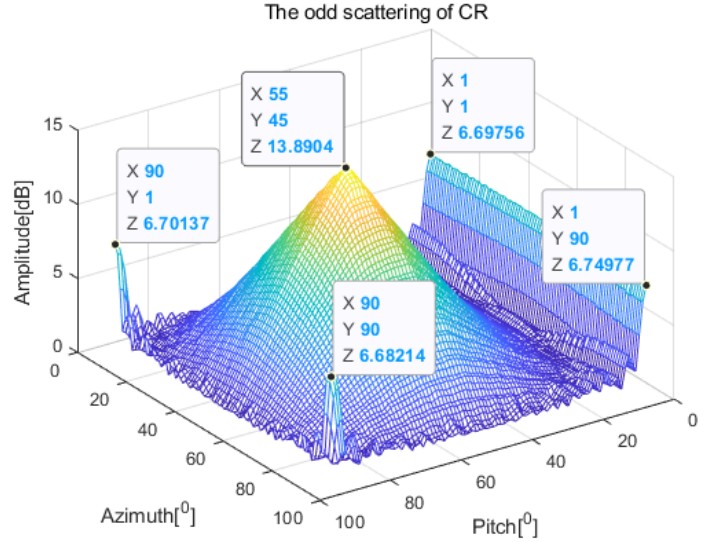

**Figure A2.** The odd scattering of CR.

Figure A3 presents that how the coefficient of CR even scattering changes with pitch and azimuth angles. The DR $S_{DR12}$ is the largest among the symmetrical parallel axes of planes 1 and 2, corresponding to the area $\varphi = 0°$ and $\theta = 45°$ in Figure A3. The DR $S_{DR23}$ is the largest among the symmetrical parallel axes of planes 2 and 3, corresponding to the area $\varphi = 45°$ and $\theta = 90°$. The DR $S_{DR31}$ is the largest among the symmetrical parallel axis of planes 3 and 1, corresponding to the area $\varphi = 90°$ and $\theta = 45°$. For the even scattering containing $S_{DR12}$, $S_{DR23}$, and $S_{DR31}$ at the same time, the even scattering is a single-peak curve when the azimuth or pitch angle changes within $45°$, which is a minimum of $[45°, 90°]$, as shown in Figure A3.

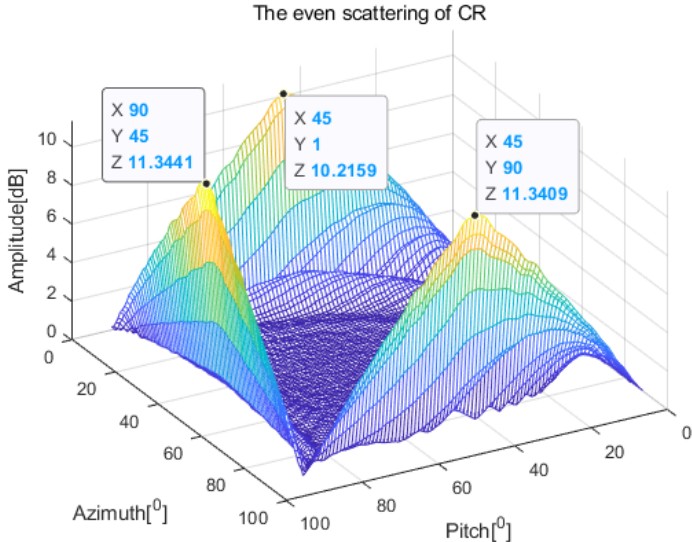

**Figure A3.** The even scattering of CR.

Combining the changing rule of odd scattering and even scattering, we find that the MSP coefficient of CR is a single-peak curve, when the azimuth or pitch angle of radar incident wave changes monotonically within 35°.

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
