# Peer review of "A Ship Discrimination Method Based on High-Frequency Electromagnetic Theory"

_remotesensing, doi:10.3390/rs14163893_

Round 1

Reviewer 1 Report

In their paper, the authors introduce an efficient approach for the discrimination of ship from the clutter. The problem is mainly the discrimination  of corner reflectors (CR) and ships. The CR diffraction is an important problem because its RCS is important and contributes to bad interpretations in ship detection. The different approaches used for such a problem are quoted and clearly referenced in the introduction, their advantages and drawbacks are described. The first par is dedicated to the presentation of an interesting model of the CR. The theoretical part based on high frequency approach is well developed and the decomposition with different number of reflections allows to give analytical formulations. Then the approach based on the decomposition on the main polarization is presented. The MAE calculation is made to quantify the performances of the approach.  Some interesting examples illustrate the approach and validate its performance. 

Some typing error have to be corrected:

Line 97:  developed

Line 104 :  comparison

Line 124 : polarization

This paper suggest some questions to the reviewer:

-       The formula (53) gives a threshold level Tvalue which seems to be adjusted in function of the problem, polarization,.. are there more precise criteria ?

-       Concerning the criteria of discretization of the triangles, this  notion appears in the part 5 dedicated to simulation. Is the lambda/10 criterion necessary in all cases, are there any constrains on the shape of the discretized triangles (two long edges and the third short for example) ?

-       Can the authors explain why the error obtained for the CR decrease strongly for SCR > 5 dB compared to the other more classical  approaches  ?

As a conclusion, this paper show an interesting discrimination approach, that gives an important discrimination value (95%) with a very low false alarm probability (4%).

 The paper clearly describe the approach and then merit a publication cosidering the above remarks.

Author Response

Dear expert,

 Thank for your patience and help, we have carefully revised the paper according to the all comments of yours. The revised content is shown in 'Respond reviewer.PDF'.

Thanks again for your comments.

Sincerely,

Dr. Yang

Reviewer 2 Report

The corner reflector (CR) can successfully deceive radar by its strong RCS for protecting ship. The authors proposed a novel discrimination method between CR and ship based on 3D characteristics via deducing the basic scattering of CR using their decomposition, and in such way solving Krogager decomposition method that has large errors in clutter. Authors analyzed the time-spatial-polarization (TSP) joint domains characteristics of radar based on the 3D proportional guidance. The comparison of novel method using the fully polarimetric data of Feko software confirmed 95% discrimination probability, and 4.1% false alarm probability.

Comments:

1  1)   In opinion of this reviewer, the authors should explain the CR discrimination method (Fig. 8) in the form of pseudocode algorithm where all used operations can be presented in accordance with the theory proposed in previous sections. This can help for potential reader bin better understanding of novel method.

2   2) The authors introduce the judgment threshold Tvalue (eq. 53). The authors wrote: “this paper analyzes the judgment thresholds under different discrimination probabilities and different false alarm probabilities by means of a large number of simulations. The detail will be explained in Section V.C.”, but this reviewer did not find explicitly clear explications and suggests better justifying this selected value.

        3) This  reviewer thinks that for comparison of the proposed method with two other ones used by authors, it should be presented much more details of the Monte Carlo simulations analysis. The authors only wrote (page 22): “By performing 10000 Monte Carlo simulations, the discrimination probability and false alarm probability of three methods are exhibited in Table. III. the HRRP’s method, its results are very susceptible to the arrangement of CR and type of ship, thus having the highest false alarm probability. For the polarization’s method, it only utilizes the polarization information at an independent moment, which has an error when the incident wave is far from the LOS symmetry axis.”

Author Response

(The authors gave the same response as above.)

Reviewer 3 Report

Your paper could be made shorter. For example you can compress some maths or reference it in a straightforward way. Also the inroduction could be more compact although I admire your set of references.

It would be better to denote Appendix with letter A or without a letter instead of letter G. In the appendix figures are denoted with letter A although they follow numbering from the main part of the paper.

1. The possibility to recognise objects on sea with high certainty and possibility of objects classification.

2. The topic is relevant in the field.

3. The authors claim that discrimination method based on the time-spatial-polarization (TSP) joint domains improves the accuracy of ship discrimination.

I agree with authors. Their method is an interesting step up in the theory and methods of ship discrimination.

4. The authors could provide more practical examples  (e.g. HRRP characteristics of different ships) and as I have written in the review the theory could be compressed.

5. The conclusions are consistent with the theory and results presented in the paper.

6. The references seem to be appropriate although some radar oriented researches would probably provide wider scope of references in the area of signal processing.

Author Response

(The authors gave the same response as above.)

Round 2

Reviewer 2 Report

The authors have attended all comments of this reviewer.